# Assumption violations in causal discovery and the robustness of score matching

**Francesco Montagna**
MaLGa, Università di Genova

**Atalanti A. Mastakouri**
AWS

**Elias Eulig**
German Cancer Research Center (DKFZ)

**Nicoletta Noceti**
MaLGa, Università di Genova

**Lorenzo Rosasco**
MaLGa, Università di Genova
MIT, CBMM
Istituto Italiano di Tecnologia

**Dominik Janzing**
AWS

**Bryon Aragam**
University of Chicago

**Francesco Locatello**
Institute of Science and Technology Austria (ISTA)

## Abstract

When domain knowledge is limited and experimentation is restricted by ethical, financial, or time constraints, practitioners turn to observational causal discovery methods to recover the causal structure, exploiting the statistical properties of their data. Because causal discovery without further assumptions is an ill-posed problem, each algorithm comes with its own set of usually untestable assumptions, some of which are hard to meet in real datasets. Motivated by these considerations, this paper extensively benchmarks the empirical performance of recent causal discovery methods on observational *iid* data generated under different background conditions, allowing for violations of the critical assumptions required by each selected approach. Our experimental findings show that score matching-based methods demonstrate surprising performance in the false positive and false negative rate of the inferred graph in these challenging scenarios, and we provide theoretical insights into their performance. This work is also the first effort to benchmark the stability of causal discovery algorithms with respect to the values of their hyperparameters. Finally, we hope this paper will set a new standard for the evaluation of causal discovery methods and can serve as an accessible entry point for practitioners interested in the field, highlighting the empirical implications of different algorithm choices.

## 1 Introduction

The ability to infer causal relationships from observational data, instead of simple statistical associations, is crucial to answer interventional and counterfactual queries without direct manipulation of a system [1, 2, 3, 4]. The challenge of drawing causal conclusions from pure observations lies in the modeling assumptions on the data, which are often impossible to verify. Methods based on conditional independence testing (e.g. PC, FCI and their variations [4, 5, 6]) require *faithfulness* of the distribution [1, 2, 4, 7] to the causal graph, which formalizes the intuition that causal relations manifest themselves in the form of statistical dependencies among the variables. The assumption of *causal sufficiency* (i.e. the absence of unobserved confounders [8]) is a common requirement for

37th Conference on Neural Information Processing Systems (NeurIPS 2023).

causal discovery [4, 6, 9, 10, 11, 12], which allows interpreting associations in the data as causal relationships. These strong conditions are arguably necessary but nevertheless hard or impossible to verify, and posit an entry barrier to the unobscured application of causal analysis in general settings. In addition to that, structure identifiability results define limitations on the parts of the causal graph that can be inferred from pure observations [1, 10, 13]. Traditional causal discovery methods (e.g. PC, FCI, GES [4, 9]) are limited to the inference of the Markov Equivalence Class of the ground truth graph [14], while additional assumptions on the structural equations generating effects from the cause ensure identifiability of a unique Directed Acyclic Graph (DAG) from observational data. In particular, restrictions on the class of functions generating the data (linear or not) and on the distribution of the noise terms (i.e. additive noise models assumed in LINGAM and more) characterizing their non-deterministic relationships are necessary in order to infer causal directions [10, 11, 12, 13]. Requirements on the data collection process are also needed: although an error-free measurement model is commonly assumed, it has been a recent subject of interest that measurement error in the observed values of the variables can greatly change the output of various causal discovery methods [15, 16].

Real data hardly satisfy all of these assumptions at once, and it is often the case that these are impossible to verify, which calls for algorithms that demonstrate a certain degree of robustness with respect to violations of the model hypothesis. Previous work from Heinze-Deml et al. [17] investigates the boundaries of robust graph inference under model misspecifications, on Structural Causal Models (SCM) with linear functional mechanisms. Mooij et al. [18] benchmark considers the case of additive noise models with nonlinear mechanisms, but only for datasets with two variables. Singh et al. [19] presents an empirical evaluation limited to methods whose output is a Markov Equivalence Class. Glymour et al. [14] review some of the existing approaches with particular attention to their required assumptions, but without experimental support to their analysis. Our paper presents an extensive empirical study that evaluates the performance of classical and recent causal discovery methods on observational datasets generated from *iid* distributions under diverse background conditions. Notably, the effects of these conditions on most of the methods included in our benchmark have not been previously investigated. We compare causal discovery algorithms from the constraint and score-based literature, as well as methods based on restricted functional causal models of the family of additive nonlinear models [11, 13, 20]. These include a recent class of methods deriving connections between the score matching [21, 22] with the structure of the causal graph [23, 24, 25]. Algorithms that focus on *sequential* data [26, 27, 28, 29, 30, 31, 32, 33, 34, 35, 36, 37] are beyond the scope of this paper's benchmarking. Finally, we propose an experimental analysis of the stability of the benchmarked approaches with respect to the choices of their hyperparameters, which is the first effort of this type in the literature.

We summarise the contributions of our paper as follows:

- We investigate the performance of current causal discovery methods in a large-scale experimental study on datasets generated under different background conditions with violations of the required background assumptions. Our experimental protocol consists of more than 2M experiments with 11 different causal discovery methods on more than 60000 datasets synthetically generated.

- We release the code for the generation of the synthetic data and a Python implementation of six causal discovery algorithms with a shared API. With this contribution, we aim at facilitating the benchmarking of future work in causal discovery on challenging scenarios, and the comparison with the most prominent existing baselines.

- We analyze our experimental results, and present theoretical insights on why score matching-based approaches show better robustness in the setting where assumptions on the data may be violated, compared to the other methods. Based on our empirical evidence, we suggest a new research direction focused on understanding the role of the statistical estimation algorithms applied for causal inference, and the connection of their inductive biases with good empirical performance.

## 2   The causal model

In this section, we define the problem of causal discovery, with a brief introduction to the formalism of Structural Causal Models (SCMs). Then we provide an overview of SCMs for which sufficient conditions for the *identifiability* of the causal graph from observational data are known.

## 2.1 Problem definition

A Structural Causal Model $\mathcal{M}$ is defined by the set of *endogenous* variables $\mathbf{X} \in \mathbb{R}^d$, vertices of the causal graph $\mathcal{G}$ that we want to identify, the *exogenous* noise terms $\mathbf{U} \in \mathbb{R}^d$ distributed according to $p_{\mathbf{U}}$, as well as the functional mechanisms $\mathcal{F} = (f_1, \ldots, f_d)$, assigning the value of the variables $X_1, \ldots, X_d$ as a deterministic function of their causes and of some random disturbance.

Each variable $X_i$ is defined by a structural equation:

$$X_i := f_i(\mathrm{PA}_i, U_i), \ \forall i = 1, \ldots, d, \tag{1}$$

where $\mathrm{PA}_i \subset \mathbf{X}$ is the set of parents of $X_i$ in the causal graph $\mathcal{G}$, and denotes the set of direct causes of $X_i$. Under this model, the recursive application of Equation (1) entails a joint distribution $p_{\mathbf{X}}$, such that the Markov factorization holds:

$$p_{\mathbf{X}}(\mathbf{X}) = \prod_{i=1}^{d} p_i(X_i | \mathrm{PA}_i), \tag{2}$$

The goal of causal discovery is to infer the causal graph underlying $\mathbf{X}$ from a set of $n$ observations sampled from $p_{\mathbf{X}}$.

## 2.2 Identifiable models

In order to identify the causal graph of $\mathbf{X} \in \mathbb{R}^d$ from purely observational data, further assumptions on the functional mechanisms in $\mathcal{F}$ and on the joint distribution $p_{\mathbf{U}}$ of model (1) are needed. Intuitively, having one condition between nonlinearity of the causal mechanisms and non-Gaussianity of the noise terms is necessary to ensure the identifiability of the causal structure. Additionally, we consider causal *sufficiency* (Appendix A.3) of the model to be satisfied, unless differently specified.

**Linear Non-Gaussian Model (LINGAM).** A linear SCM is defined by the system of structural equations

$$\mathbf{X} = \mathbf{B}\mathbf{X} + \mathbf{U}. \tag{3}$$

$\mathbf{B} \in \mathbb{R}^{d \times d}$ is the matrix of the coefficients that define $X_i$ as a linear combination of its parents and the disturbance $U_i$. Under the assumption of non-Gaussian distribution of the noise terms, the model is identifiable. This SCM is known as the LiNGAM (Linear Non-Gaussian Acyclic Model) [12].

**Additive Noise Model.** An Additive Noise Model (ANM) [11, 13] is defined by Equation (1) when it represents the causal effects with nonlinear functional mechanisms and additive noise terms:

$$X_i := f_i(\mathrm{PA}_i) + U_i, \ \forall i = 1, \ldots, d, \tag{4}$$

with $f_i$ nonlinear. Additional conditions on the class $\mathcal{F}$ of functional mechanisms and on the joint distribution of the noise terms are needed to ensure identifiability [13]. In the remainder of the paper, we assume these to hold when referring to ANMs.

**Post NonLinear Model.** The most general model for which sufficient conditions for the identifiability of the graph are known is the Post NonLinear model (PNL) [10]. In this setting the structural equation (1) can be written as:

$$X_i := g_i(f_i(\mathrm{PA}_i) + U_i), \ \forall i = 1, \ldots, d, \tag{5}$$

where both $g_i$ and $f_i$ are nonlinear functions and $g_i$ is invertible. As for ANMs, we consider identifiability conditions defined in Zhang and Hyvärinen [10] to be satisfied in the rest of the paper.

# 3 Experimental design

In this section, we describe the experimental design choices regarding the generation of the synthetic datasets, the evaluated methods, and the selected metrics.

## 3.1 Datasets

The challenge of causal structure learning lies in the modeling assumptions of the data, which are often untestable. Our aim is to investigate the performance of existing causal discovery methods in the setting where these assumptions are violated. To this end, we generate synthetic datasets under

| | PC | FCI | GES | DirectLiNGAM | RESIT | CAM | SCORE | DAS | NoGAM | DiffAN | GraN-DAG |
|---|---|---|---|---|---|---|---|---|---|---|---|
| Gaussian noise | ✓ | ✓ | ✓ | ✗ | ✓ | ✓ | ✓ | ✓ | ✓ | ✓ | ✓ |
| Non-Gaussian noise* | ✓ | ✓ | ✗ | ✓ | ✓ | ✗ | ✗ | ✗ | ✓ | ✗ | ✗ |
| Linear mechanisms | ✓ | ✓ | ✓ | ✓ | ✗ | ✗ | ✗ | ✗ | ✗ | ✗ | ✗ |
| Nonlinear mechanisms | ✓ | ✓ | ✓ | ✗ | ✓ | ✓ | ✓ | ✓ | ✓ | ✓ | ✓ |
| Unfaithful distribution | ✗ | ✗ | ✗ | ✓ | ✓ | ✓ | ✓ | ✓ | ✓ | ✓ | ✓ |
| Confounding effects | ✗ | ✓ | ✗ | ✗ | ✗ | ✗ | ✗ | ✗ | ✗ | ✗ | ✗ |
| Measure errors | ✗ | ✗ | ✗ | ✗ | ✗ | ✗ | ✗ | ✗ | ✗ | ✗ | ✗ |
| Output | CPDAG | PAG | CPDAG | DAG | DAG | DAG | DAG | DAG | DAG | DAG | DAG |

* GraN-DAG and GES optimize the Gaussian likelihood.

Table 1: Summary of the methods assumptions and their output graph. The content of the cells denotes whether the method supports (✓) or not (✗) the condition specified in the corresponding row.

diverse background conditions, defined by modeling assumptions that do not match the working hypothesis of the evaluated methods.

**Vanilla model.** First, we specify an additive noise model with variables generated according to the structural equation (4). The exogenous terms follow a Gaussian distribution $U_i \sim \mathcal{N}(0, \sigma_i)$ with variance $\sigma_i \sim U(0.5, 1.0)$ uniformly sampled. We generate the nonlinear mechanisms $f_i$ by sampling Gaussian processes with a unit bandwidth RBF kernel (Appendix B.1). We refer to this model as the *vanilla* scenario, as it is at one time both identifiable and compliant with the assumptions of the majority of the benchmarked methods (see Table 1).

### 3.1.1 Misspecified scenarios

We define additional scenarios such that each specified model targets a specific assumption violation with respect to the vanilla conditions.

**Confounded model.** Let $\mathbf{Z} \in \mathbb{R}^d$ be a set of latent common causes. For each pair of distinct nodes $X_i$ and $X_j$, we sample a Bernoulli random variable $C_{ij} \sim Bernoulli(\rho)$ such that $C_{ij} = 1$ implies a confounding effect between $X_i$ and $X_j$. The index $k$ of the confounder $Z_k$ is assigned at random. The parameter $\rho \in \{0.1, 0.2\}$ determines the amount of confounded pairs in the graph.

**Measurement error model.** Measurement errors in the process that generates the data are regarded as a source of mistakes for the inference of the causal graph [15, 16]. In order to account for potential errors induced by the measurements, we specify a model in which the observed variables are:

$$\tilde{X}_i := X_i + \epsilon_i, \forall i = 1, \dots, d, \tag{6}$$

a noisy version of the $X_i$'s generated by the ANM of Equation (4). The $\epsilon_i$ disturbances are independent Gaussian random variables centered at zero, whose variance is parametrized by the inverse signal-to-noise ratio $\gamma_i := \frac{\text{Var}(\epsilon_i)}{\text{Var}(X_i)}$. Given that the total variance of $\tilde{X}_i$ is $\text{Var}(\tilde{X}_i) = \text{Var}(X_i) + \text{Var}(\epsilon_i)$, $\gamma_i$ controls the amount of variance in the observations that is explained by the error in the measurement. Each dataset with measurement error is parametrized with $\gamma \in \{0.2, 0.4, 0.6, 0.8\}$, shared by all the $\epsilon_i$.

**Unfaithful model.** To model violations of the *faithfulness* assumption (Appendix A.2), we tune the causal mechanisms of Equation (4) such that we induce direct cancellation of causal effects between some variables. In particular, for each triplet $X_i \to X_k \leftarrow X_j \leftarrow X_i$ in the graph, causal mechanisms are adjusted such that cancellation of the causal effect $X_i \to X_k$ occurs (for implementation details, see Appendix B.4). This is a partial model of unfaithfulness, as it only covers a limited subset of the scenarios under which unfaithful path canceling might occur, and must be viewed in the light that there is no established procedure to enforce unfaithful conditional independencies in the case of ANM with nonlinear relationships.

**Autoregressive model.** In order to simulate violations of the *iid* distribution of the data, we model observations as a stochastic process where each sample is indexed by time. In particular, we define the structural equations generating the data as:

$$X_i(t) := \alpha X_i(t-1) + f_i(\text{PA}_i(t)) + U_i, \quad t = 1, 2, 3, \dots \quad \alpha \in \mathbb{R}. \tag{7}$$

Autoregressive effects are modeled with a time lag $l = 1$, whereas at $t = 0$ we define $X_i(0)$ with Equation (4). The ground truth is the graph whose edges represent the connections between parents $\text{PA}_i(0)$ and their direct effect $X_i(0)$.

**Post NonLinear model**. We replace nonlinear causal mechanisms of the additive noise models (4) with the structural equations defined in the PNL model (5). We select the post nonlinear function $g_i$ such that $g_i(x) = x^3, x \in \mathbb{R}, \forall i = 1, \ldots, d$.

**LiNGAM model**. We define a model with the linear system of structural equations (3). The non-Gaussian distribution of the noise terms is defined as a nonlinear transformation of a standard normal random variable (see Appendix B.2), and the linear mechanisms are simulated by sampling the weighting coefficients of the parents of a node in the interval $[-1, -0.05] \cup [0.05, 1]$.

### 3.1.2 Data generation

For each specified model, we generate datasets that differ under the following characteristics: number of nodes $d \in \{5, 10, 20, 50\}$, number of samples $n \in \{100, 1000\}$ and density of edges $p \in \{\text{sparse}, \text{dense}\}$. We sample the ground truth causal structures according to different algorithms for random graph generation. In line with previous causal discovery literature [23, 24, 25, 38, 39] we generate Erdos-Renyi (ER) [40] and Scale-free (SF) graphs [41]. Furthermore, we consider Gaussian Random Partitions (GRP) [42] and Fully Connected graphs (FC) (see Appendix B.3). By considering all the combinations of the number of nodes, number of samples, admitted edge densities, and algorithms for structure generation, we define a cartesian product with all the graph configurations of interest. For each of such configurations and for each modeling scenario, we generate a dataset $\mathcal{D}$ and its ground truth $\mathcal{G}$ with 20 different random seeds. Details on the generated data can be found in Appendix B.5.

### 3.2 Methods

We consider 11 different algorithms and a random baseline spanning across the main families of causal discovery approaches: constraint and score-based methods, and methods defined under restrictions on the structural causal equations. In the main text, we provide a detailed overview of the methods most relevant for the discussion of our key experimental findings. The remaining approaches are described in further detail in the Appendix C. Table 1 summarizes the algorithms' assumptions and the output object of their inference procedure.

**Method outputs.** Causal discovery algorithms output different graphical objects based on their underlying assumptions. If identifiability is not implied by the model requirements but *faithfulness* of the distribution is satisfied, one can instead recover the Markov equivalence class of the ground truth graph, that is, the set of DAGs sharing the same conditional independencies. This is represented by a complete partially directed acyclic graph (CPDAG), where undirected edges $X_i — X_j$ are meant to encode conditional dependence between the variables, but uncertainty in the edge orientation. If a method can identify a directed acyclic graph $\mathcal{G} = (\mathbf{X}, \mathcal{E})$, one can define a partial ordering of the nodes $\pi = \{X_{\pi_1}, \ldots, X_{\pi_d}\}, \pi_i \in \{1, \ldots, d\}$, such that whenever we have $X_{\pi_i} \rightarrow X_{\pi_j} \in \mathcal{E}$, then $X_{\pi_i} \prec X_{\pi_j}$ ($X_{\pi_j}$ is a *successor* of $X_{\pi_i}$ in the ordering) [43]. The permutation $\pi$ is known as the *topological order* of $\mathcal{G}$, and allows to disambiguate the direction of the edges in the graph. A topological order can be encoded in a fully connected DAG with edges $\mathcal{E}_\pi = \{X_{\pi_i} \rightarrow X_{\pi_j} : X_{\pi_i} \prec X_{\pi_j}, \forall i, j = 1, \ldots, d\}$, obtained connecting all nodes in the ordering $\pi$ with their successors.

**Methods summary.** A summary of all the methods included in the benchmark and their required assumptions is presented in Table 1. PC [4] and GES [9] are limited to identifying the Markov equivalence class of the DAG. DirectLiNGAM [44] is designed for inference on data generated by a linear non-Gaussian model whereas SCORE [23], NoGAM [25], DiffAN [45], DAS [24], RESIT [13], GraN-DAG [38] and CAM [46], are meant for inference on additive noise models: these methods perform inference in a two steps procedure, first identifying a topological ordering of the graph, and then selecting edges between those admitted by the inferred causal order. To enable fair comparison in our experiments, all methods (with the exception of DirectLiNGAM) are implemented with the same algorithm for edge detection, consisting of variable selection with sparse regression. This pruning strategy is known as *CAM-pruning*, being originally proposed in CAM paper [46]. A detailed discussion of all the methods in the benchmark is presented in Appendix C. In the Appendix L we consider experiments on FCI [4], which are not reported in the main text since we did not find metrics for a straightforward comparison of its output partial ancestral graph (PAG [47]) with CPDAGs and DAGs.

**Selected metrics** To evaluate the output graphs we use the false positive and false negative rates, and the F1 score (details in the Appendix D). In the case of directed edges inferred with reversed direction, we count this error as a false negative. For methods that output a CPDAG with undirected edges, we evaluate them favorably by assuming correct orientation whenever possible, similar to Zheng et al. [39, 48]. For the methods whose output also includes an estimate $\hat{\pi}$ of the topological order, we define the false negative rate of the fully connected DAG with edges $\mathcal{E}_{\hat{\pi}} = \{X_{\hat{\pi}_i} \to X_{\hat{\pi}_j} : X_{\hat{\pi}_i} \prec X_{\hat{\pi}_j}, \forall i, j = 1, \ldots, d\}$, denoted as FNR-$\hat{\pi}$. If $\hat{\pi}$ is correct with respect to the ground truth graph, then FNR-$\hat{\pi} = 0$.

This choice of metrics reflects the implementation of most of the algorithms involved in the benchmark, which separates the topological ordering step from the actual edge selection. In particular, given that the majority of the methods share the same pruning procedure after the inference of the order, we expect that differences in the performance will be mostly observed in the FNR-$\hat{\pi}$ score.

### 3.2.1  Deepdive on SCORE, NoGAM and DiffAN

In this section, we review a recent class of causal discovery algorithms, that derive constraints on the score function $\nabla \log p(\mathbf{X})$ that uniquely identifies the directed causal graph of an additive noise model. Identifiability assumptions provide sufficient conditions to map a joint distribution $p_{\mathbf{X}}$ to the unique causal DAG $\mathcal{G}$ induced by the underlying SCM. Applying the logarithm to the Markov factorization of the distribution of Equation (2), we observe that $\log p_{\mathbf{X}}(\mathbf{X}) = \sum_i^d \log p(X_i \mid \mathrm{PA}_i)$. By inspection of the gradient vector $\nabla \log p_{\mathbf{X}}(\mathbf{X})$, it is possible to derive constraints mapping the score function to the causal graph of an ANM. Given a node $X_i$ in the graph, its corresponding score entry is defined as:

$$s_i(\mathbf{X}) := \partial_{x_i} \log p_{\mathbf{X}}(\mathbf{X}) = \partial_{x_i} \log p_i(X_i \mid \mathrm{PA}_i) + \sum_{k \in \mathrm{CH}_i} \partial_{x_i} \log p_k(X_k \mid \mathrm{PA}_k). \tag{8}$$

Instead, the rate of change of the log-likelihood over a leaf node $X_l$ with the set of children $\mathrm{CH}_l = \emptyset$ is:

$$s_l(\mathbf{X}) := \partial_{x_l} \log p_{\mathbf{X}}(\mathbf{X}) = \partial_{x_l} \log p_l(X_l | \mathrm{PA}_l). \tag{9}$$

We see that, for a leaf node, the summation over the set of children of Equation (8) vanishes. Intuitively, being able to capture this asymmetry in the score entries enables the identification of the topological order of the causal graph.

**SCORE.** The SCORE algorithm [23] identifies the topological order of ANMs with Gaussian noise terms by iteratively finding leaf nodes as the $\mathrm{argmin}_i \mathrm{Var}[\partial_{x_i} s_i(\mathbf{X})]$, given that the following holds:

$$\mathrm{Var}\left[\partial_{x_i} s_i(\mathbf{X})\right] = 0 \iff X_i \text{ is a leaf}, \ \forall i = 1, \ldots, d. \tag{10}$$

**NoGAM.** The NoGAM [24] algorithm generalizes the ideas of SCORE on additive noise models with an arbitrary distribution of the noise terms. After some manipulations, it can be shown that for a leaf node $X_l$ the score entry of Equation (9) satisfies

$$s_l(\mathbf{X}) = \partial_{U_l} \log p_l(U_l), \tag{11}$$

such that one could learn a consistent estimator of $s_l$ taking as input the exogenous variable $U_l$. For an ANM, the authors of NoGAM show that the noise term of a leaf is equivalent to the residual defined as:

$$R_i := X_i - \mathbf{E}\left[X_i \mid \mathbf{X} \setminus \{X_i\}\right], \forall i = 1, \ldots, d. \tag{12}$$

Then, by replacing $U_l$ with $R_l$ in Equation (11), it is possible to find a consistent approximator of the score of a leaf using $R_l$ as the predictor. Formally:

$$\mathbf{E}\left[\left(\mathbf{E}\left[s_i(\mathbf{X}) \mid R_i\right] - s_i(\mathbf{X})\right)^2\right] = 0 \iff X_i \text{ is a leaf}, \tag{13}$$

which identifies a leaf node as the $\mathrm{argmin}$ of the vector of the mean squared errors of the regression of the score entries $s_i(\mathbf{X})$ on the corresponding residuals $R_i$, for all $i = 1, \ldots, d$.

**Connection of NoGAM with the post nonlinear model.** It is interesting to notice that, similarly to Equation (11) for additive noise models, the score of a leaf $X_l$ generated by a PNL model can be defined as a function of the disturbance $U_l$.

**Proposition 1.** *Let $\mathbf{X} \in \mathbb{R}^d$ be generated according to the post nonlinear model* (5). *Then, the score function of a leaf node $X_l$ satisfies $s_l(\mathbf{X}) = \partial_l \log p_l(U_l)$.*

This result suggests a connection with the NoGAM sorting criterion: indeed, one could hope to identify leaf nodes in the graph by consistent estimation of the score of a leaf from residuals equivalent to the noise terms. A more detailed discussion with the proof of Proposition 1 is presented in Appendix E.

**DAS.** The DAS algorithm (acronym for Discovery At Scale, [24]) identifies the topological ordering with the same procedure defined in SCORE, while the two methods differ in the way they find edges in the graph. DAS edge selection procedure exploits the information in the non-diagonal entries of the Jacobian of the score. In particular, for ANM with Gaussian noise terms, it can be shown that:

$$\mathbf{E}\left[\left|\partial_{x_j} s_l(\mathbf{X})\right|\right] \neq 0 \iff X_j \in \mathrm{PA}_l(\mathbf{X}), \ \forall j \in \{1, \ldots, d\} \setminus \{l\} \tag{14}$$

Exploiting Equation 14 to define the inference rule for the edge selection in DAS provides a significant computational advantage with respect to SCORE, reducing the time complexity in the number of nodes from cubic to quadratic.

**DiffAN.** DiffAN [45] method finds the topological ordering of a DAG exploiting the same criterion of Equation (10) of SCORE: the difference is in that it estimates the score function with probabilistic diffusion models, whereas SCORE, NoGAM, and DAS [24] rely on score matching estimation [21, 22].

## 4 Key experimental results and analysis

In this section we present our experimental findings on datasets generated according to the misspecified models of Section 3.1.1, with theoretical insights into the performance of score matching-based approaches. We draw our conclusions by comparing the methods' performance against their accuracy in the vanilla scenario and against a random baseline [1] (defined in Appendix C.10). The results are discussed on datasets of size 1000 for Erdos-Renyi dense graphs with 20 nodes (*ER-20 dense*), and can be generalized to different size and sparsity configurations. Due to space constraints, we include the plots only for the F1 score and FNR-$\hat{\pi}$, whereas the false negative and false positive rates are discussed in Appendix I. In order to provide statistical significance to our conclusions, we repeat the experiments on each scenario over 20 datasets generated with different random seeds. To enable a fair comparison between the methods, we fix their hyperparameters to their optimal value with respect to each specific dataset, in the case where these can not be tuned without having access to the ground truth (see Appendix G for a discussion on the tuning of GraNDAG and DiffAN learning hyperparameters). In the Appendix H we analyze the stability of the benchmarked methods with respect to different values of their hyperparameters.

### 4.1 Can current methods infer causality when assumptions on the data are violated?

Our experimental findings suggest that score matching-based algorithms can robustly infer part of the causal information even in the case of misspecified ground truth data generation.

**Post nonlinear model.** Figure 1a (right) illustrates the accuracy of topological order estimates on post nonlinear model data. Among the selected methods, NoGAM shows better ability to generalize its performance to this scenario, with FNR-$\hat{\pi}$ error rate significantly lower than the random baseline. Interestingly, we can interpret this observation in the light of Proposition 1, which defines the score of a leaf in the PNL model: our result indeed suggests that, similarly to the case of an additive noise model, it is possible to learn a consistent approximator of the score of a leaf $X_l$ from the exogenous variable $U_l$ of a post nonlinear model. Notably, we also observe that RESIT order accuracy is better in the PNL scenario than in the vanilla case: Zhang and Hyvärinen [10] show that testing for independent residuals identifies the direction of causal relationships also under the PNL model.

**LiNGAM model.** Figure 1b (right) shows that NoGAM can infer the causal order with remarkable accuracy in the case of ground truth data generated by a linear non-gaussian additive model. Together with our observations on the post nonlinear model, our empirical evidence corroborates the idea that the NoGAM algorithm is surprisingly robust with respect to the misspecification of the causal mechanisms. Notably, none of the other methods can infer the ordering with accuracy significantly better than the random baseline. This could lead to decreased performance in the realistic setting of mixed linear and nonlinear mechanisms. However, the F1 score in Figure 1b (left) shows that CAM-pruning is still able to correctly infer edges in the graph when these are admitted by the identified causal order. We note that, given that we observed high *varsortability*[2][50] for this model, we display results on data standardized dividing by their empirical variance.

---

[1]We use the `https://github.com/cdt15/lingam` implementations of RESIT and DirectLiNGAM, and the DoDiscover implementations of PC, GES, and FCI. For the remaining methods, we consider the GitHub official repositories of their papers and custom implementations.

[2]*Varsortability* of a dataset denotes partial agreement between the ordering induced by the values of marginal variance of the observed variables and the causal ordering of the underlying graphical model. Note that other simple baselines can still recover the causal order after rescaling of the data [49].

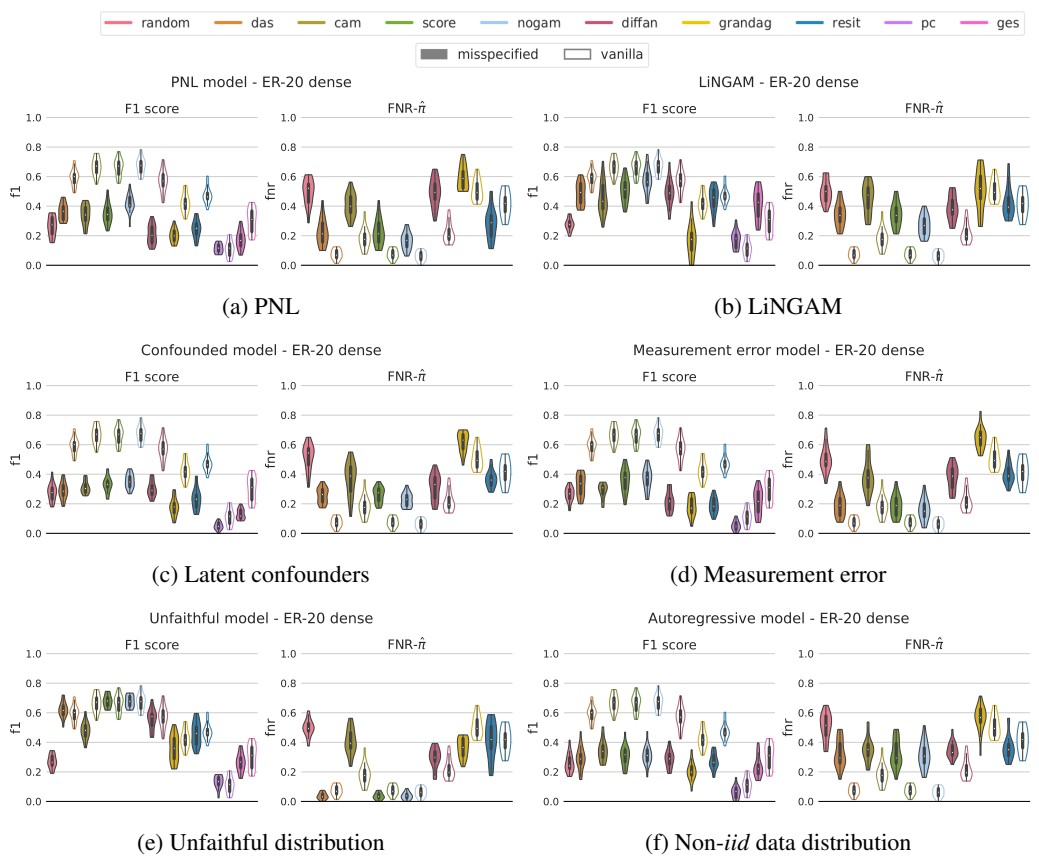

Figure 1: Experimental results on the misspecified scenarios. For each method, we also display the violin plot of its performance on the *vanilla* scenario with transparent color. F1 score (the higher the better) and FNR-$\hat{\pi}$ (the lower the better) are evaluated over 20 seeds on Erdos-Renyi dense graphs with 20 nodes (ER-20 dense). FNR-$\hat{\pi}$ is not computed for GES and PC, methods whose output is a CPDAG. Note that DirectLiNGAM performance is reported in Appendix I.2, on data under non-Gaussian distribution of the noise terms.

**Confounded model.** Spurious correlations, that occur when the causal sufficiency is violated, can not be handled by statistical tests for edge selection, as shown by the F1 score of Figure 1c (left) (the amount of confounders is parametrized by $\rho = 0.2$). In this case, we are also interested to see whether the presence of latent confounders can disrupt the inference of the topological ordering when the observed variables have a non-spurious connection in the causal graph. Figure 1c (right) indicates that the score matching-based approaches SCORE, DAS, and NoGAM can still be exploited to find a reliable ordering, while other methods fail to do so.

**Measurement error.** Given data generated under the model of Equation (6), we observe convergence in distribution $p(\tilde{X}_i \mid \mathrm{PA}_i) \xrightarrow{d} p(X_i \mid \mathrm{PA}_i)$ for $\gamma \to 0$. We are then interested in the boundaries of robust performance of each method with respect to increasing values of $\gamma$. Figure (1d) (right) illustrates FNR-$\hat{\pi}$ on datasets with $\gamma = 0.8$ such that ~35% of the observed variance of each variable is due to noise in the measurements. Under these conditions, we see that score matching-based approaches display robustness in the inference of the order where all the other methods' capability is comparable to that of the random baseline with statistical significance. This is also reflected in Figure (1d) (left), where SCORE, DAS, and NoGAM are the only algorithms whose F1 score (slightly) improves compared to the random baseline.

**Unfaithful model.** Figure 1e (right) shows that the ordering procedure of several methods, in particular SCORE, DAS, NoGAM, and GraN-DAG, seems unaffected by direct cancellation of causal effects, in fact displaying a surprising decrease in the FNR-$\hat{\pi}$ performance with respect to the vanilla scenario. To understand these results, we note that under the occurrence of causal effect cancellations in the ground truth graph $\mathcal{G}$, the unfaithful model defined in Section 3.1.1 generates observations of $\mathbf{X}$ according to a graph $\tilde{\mathcal{G}}$ whose causal order agrees with that of the ground truth: it

is indeed immediate to see that the causal order of $X_i \to X_k \leftarrow X_j \leftarrow X_i$ also holds for the triplet $X_i \to X_j \to X_k$. Moreover, the set of edges of the graph $\tilde{\mathcal{G}}$ is sparser than that of the ground truth, due to the cancellation of causal effects. Thus, given that inference on sparser graphs is generally easier, it can positively affect the empirical performance, in line with our observations.

**Implications.** Our experimental findings show that most of the benchmarked methods significantly decrease their performance on the misspecified models. This is particularly problematic since the violations considered in this work are realistic and met on many real-world data. On the other hand, we observe surprising robustness in the inference of score matching-based methods.

### 4.1.1 Discussion on PC and GES performance

The experimental results of Figure 1 show that the F1 score of GES and PC is consistently worse than random in the setting of Erdös-Renyi dense graphs with 20 nodes. We note that this pertains specifically to large graphs with dense connections, where the accuracy of both methods significantly degrades. In the case of PC, this is in line with previous theoretical findings in the literature: Uhler et al. [7] demonstrate that large and dense graphs are characterized by many *quasi-violations* of the faithfulness assumptions, which can negatively impact the algorithm's inference ability. In Appendix M we discuss experiments on lower dimensional networks with sparse edge structures, showing that PC and GES report performance significantly better than random across different scenarios. Overall, we find that increasing the size and density of the graph negatively impacts the inference ability of PC and GES.

### 4.1.2 Discussion on score matching robustness

Our empirical findings indicate that score matching-based methods are surprisingly capable of partial recovery of the graph structure in several of the misspecified scenarios. We connect this robust performance to the decomposition properties of the score function defined in Equations (8) and (9). In particular, we argue that the common factor that enables leaf node identification in NoGAM and SCORE is that the score entry of a leaf is characterized by a smaller magnitude, compared to the score associated with a node that has children in the graph. To explain what we mean by this, we define a simple condition under which it is possible to identify leaf nodes and the causal order of the graph from the variance of the entries of the score function.

**Definition 1.** Let $\mathbf{X} \in \mathbb{R}^d$ be a random vector defined by a structural causal model $\mathcal{M}$ (1). Let $X_l$ be a leaf node of the causal graph $\mathcal{G}$. We say that $X_l$ is *score-identifiable* if $l = \arg\min_i \text{Var}[s_i(\mathbf{X})]$.

Moreover, we say that the model is *score-sortable* if the recursive identification of *score-identifiable* leaf nodes in the causal graph and in the subgraphs defined by removing a leaf from the set of vertices up to a source node, yields a correct causal order. SCORE, NoGAM, and DAS present results for consistent inference of the structure of an identifiable graph from properties of the score function and its second order partial derivatives. However, when these conditions are not satisfied, exploitation of *score-sortability* can heuristically estimate a causal ordering that partially agrees with the causal structure. Intuitively, the variance of the score of a non-leaf node $s_i(\mathbf{X})$ of Equation (8) is proportional to the number of children in the summation. In particular, the total variance of $s_i(\mathbf{X})$ is the sum of the marginal variances of the two terms on the RHS of Equation (8), plus their covariance. Errors in the ordering defined with *score-sortability* are induced only if the variance associated with the score of a non-leaf node can be smaller than the one relative to every leaf of the graph.

**Proposition 2.** *Let $\mathbf{X} \in \mathbb{R}^d$ be a random vector whose elements $X_i$ are defined by a structural equation model $\mathcal{M}$ (1) that satisfies score-sortabilty. Then, for each subgraph of $\mathcal{G}$ defined by recursively removing a leaf from the set of vertices up to a source node, there exists a leaf $X_l$ such that $\forall i$ index of a node:*

$$\text{Var}[\partial_{x_l} \log p_l(X_l \mid \text{PA}_l)] \le \text{Var}[\partial_{x_i} \log p_i(X_i \mid \text{PA}_i)] + \sum_{k \in \text{CH}_i} \text{Var}[\partial_{x_i} \log p_k(X_k \mid \text{PA}_k)] + C,$$

*with $C \in \mathbb{R}$ accounting for the covariance term.*

(See Appendix F for the proof.) Lemma 1 of SCORE defines a similar criterion of sortability of the causal variables on the variance of the second order partial derivatives of the log-likelihood, which is always satisfied when $\mathbf{X} \in \mathbb{R}^d$ is generated by an ANM with Gaussian distribution of

the noise terms. We can extend these considerations to the NoGAM algorithm, which identifies leaf nodes by minimizing the mean squared error of the predictions of the score entries from the residual estimators of the noise terms, as defined in Equation (13). If we consider an uninformative predictor of the score function that maps every input residual to a constant value zero, the NoGAM algorithm is equivalent to a simple *score-sortability* heuristic criterion, identifying leaf nodes as the $\arg\min_i \mathbf{E}[s_i^2(\mathbf{X})]$. In Appendix I.3 we corroborate our considerations by comparing the empirical performance of a *score-sortability* baseline with SCORE and NoGAM.

**Implications.** Score matching-based approaches SCORE, DAS, and NoGAM show empirical robustness in several scenarios included in our benchmark. We impute these results to the structure of the score function discussed in Section (3.2.1), and to the algorithmic design choices of these methods that exploit different magnitude in the score of a leaf compared to other nodes with children in the graph.

### 4.2   Is the choice of statistical estimators neutral?

In the previous section, we motivated the empirical observations on the robustness of methods based on the score function. Given that the DiffAN algorithm differs from SCORE only in the score estimation procedure (where the former applies probabilistic diffusion models in place of the score matching), we can explain the gap in performance of DiffAN with the other approaches based on the score as an effect of the different statistical estimation technique. From this observation, we suggest that score matching plays a crucial role in connecting the gradient of the log-likelihood with effective causal inference.

**Implications.** The choice of modular statistical estimator for causal inference procedures is not neutral. We argue that inductive bias in statistical estimators may be connected with good empirical performance, and we think that this potential connection should be further investigated in future works.

## 5   Conclusion

In this work we perform a large-scale empirical study on eleven causal discovery methods that provide empirical evidence on the limits of reliable causal inference when the available data violate critical algorithmic assumptions. Our experimental findings highlight that score matching-based approaches can robustly infer the causal order from data generated by misspecified models. It would be important to have procedures for edge detection that display the same properties of robustness in diverse scenarios and to have a better theoretical understanding of failure modes of CAM-pruning variable selection, given its broad use for causal discovery. Finally, we remark that this benchmarking is limited to the case of observational *iid* samples, and it would be of great practical interest to have equivalent empirical insights on the robustness of methods for causal discovery on sequential data in the setting of time series or passively observed interventions.

## 6   Acknowledgements

We thank Kun Zhang and Carl-Johann Simon-Gabriel for the insightful discussions. This work has been supported by AFOSR, grant n. FA8655-20-1-7035. FM is supported by *Programma Operativo Nazionale ricerca e innovazione 2014-2020*. FM partially contributed to this work during an internship at Amazon Web Services with FL. FL partially contributed while at AWS.

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

# Contents

## A   Assumptions connecting causal and statistical properties of the data

In this section, we describe in detail several crucial assumptions for causal discovery.

### A.1   Global Markov Property

Causal discovery from pure observations requires assumptions that connect the joint distribution of the data with their underlying causal structure. The Markov factorization of the distribution (2) allows interpreting conditional independencies of the graph $\mathcal{G}$ induced by the model $\mathcal{M}$ as conditional independencies of the joint distribution $p_{\mathbf{X}}$. This is known as the Global Markov Property of the distribution $p_{\mathbf{X}}$ with respect to the graph $\mathcal{G}$.

**Definition 2.** A distribution $p_{\mathbf{X}}$ satisfies the Global Markov Property with respect to a DAG $\mathcal{G}$ if:

$$\mathbf{X}_A \perp\!\!\!\perp_{\mathcal{G}} \mathbf{X}_B \mid \mathbf{X}_S \Rightarrow \mathbf{X}_A \perp\!\!\!\perp_{p_{\mathbf{x}}} \mathbf{X}_B \mid \mathbf{X}_S, \tag{15}$$

with $\mathbf{X}_A, \mathbf{X}_B, \mathbf{X}_S$ disjoint subsets of $\mathbf{X}$, $\perp\!\!\!\perp_{\mathcal{G}}$ denoting *d-separation* in the graph $\mathcal{G}$, and $\perp\!\!\!\perp_{p\mathbf{x}}$ denoting independency in the joint distribution $p_{\mathbf{X}}$.

## A.2 Causal faithfulness

A distribution $p_{\mathbf{X}}$ that satisfies the Global Markov Property, decomposes according to the Markov factorization of Equation (2) [51]. If the inverse holds, then we can consider the conditional independencies observed in the distribution $p_{\mathbf{X}}$ to be valid conditional independencies in the graph $\mathcal{G}$:

$$\mathbf{X}_A \perp\!\!\!\perp_{p\mathbf{x}} \mathbf{X}_B \mid \mathbf{X}_S \Rightarrow \mathbf{X}_A \perp\!\!\!\perp_{\mathcal{G}} \mathbf{X}_B \mid \mathbf{X}_S. \tag{16}$$

If (16) is satisfied, we say that $p_{\mathbf{X}}$ is *faithful* to the causal graph.

## A.3 Causal sufficiency

Another fundamental assumption is the absence of unmeasured common causes in the graph. Reichenbach principle [8] defines a connection between statistical and causal associations. The principle states that given the statistical association between two variables $X$,$Y$, then there exists a variable $Z$ that causally influences both explaining all the dependence such that conditioning on $Z$ makes them independent. *Causal sufficiency* assumes that $Z$ coincides with one between $X$ and $Y$: resorting to the model of Equation (1), this means that for each pair $X_i$,$X_j$ there are no latent common causes.

Under the assumption of *causal sufficiency* of the graph and *faithful* distribution, we can use conditional independence testing to infer the Markov Equivalence Class of the causal graph $\mathcal{G}$ from the data.

# B Details on the synthetic data generation

## B.1 Nonlinear causal mechanisms

In order to simulate nonlinear causal mechanisms of an additive noise model, we sample functions from a Gaussian process, such that $\forall i = 1, \ldots, d$, $f_i(X_{PA_i}) = \mathcal{N}(\mathbf{0}, K(X_{PA_i}, X_{PA_i}))$, a multivariate normal distribution centered at zero and with covariance matrix as the Gaussian kernel $K(X_{PA_i}, X_{PA_i})$, where $X_{PA_i}$ are the observations of the parents of the node $X_i$.

## B.2 Non-Gaussian distribution of the noise terms

We generate data with non-Gaussian noise terms as follows: for each node $i \in \{1, \ldots, d\}$, we model noise terms following a Gaussian distribution $U_i \sim \mathcal{N}(0, \sigma_i)$ with variance $\sigma_i \sim U(0.5, 1.0)$. Those noise terms are then transformed via a random nonlinear function $t$, s.t. $\tilde{U}_i = t(U_i)$. In our experiments, we sampled three different functions $t$, modeled as multilayer perceptrons (MLPs) with 100 nodes in the single hidden layer, sigmoid activation functions, and weights sampled from $U(-\alpha, \alpha), \alpha \in \{0.5, 1.5, 3.0\}$, respectively (c.f. Fig. 2).

## B.3 Algorithms for random graphs simulation

We use four random graph generation algorithms for sampling the ground truth causal structure of each dataset. In particular, we consider the Erdos-Renyi (ER) model, which allows specifying the number of nodes $d$ and the average number of connections per node $m$ (or, alternatively, the probability $p$ of connecting each pair of nodes). In ER graphs, pair of nodes have the same probability of being connected. Scale-free graphs are generated under a preferential attachment procedure [41], such that nodes with a higher degree are more likely to be connected with a new node, allowing for the presence of *hubs* (i.e. high degree nodes) in the graphs. The Gaussian Random Partition model (GRP) [42] is created by connecting $k$ subgraphs (i.e. partitions) generated by an ER model. A parameter $p_{in}$ specifies the probability of connecting a pair of nodes in the same partition, while $p_{out}$ defines the probability of connections among distinct partitions. Clusters appear when $p_{in} >> p_{out}$ (e.g. in our experiments we consider $p_{in} = 0.4$, $p_{out} = 0.05$). Finally, we consider Fully Connected graphs, generated by sampling a topological order $\pi$ and connecting all nodes in the graph to their successors with a directed edge. Given a ground truth fully connected graph, the accuracy of the inference procedure is maximally sensitive to errors in the order.

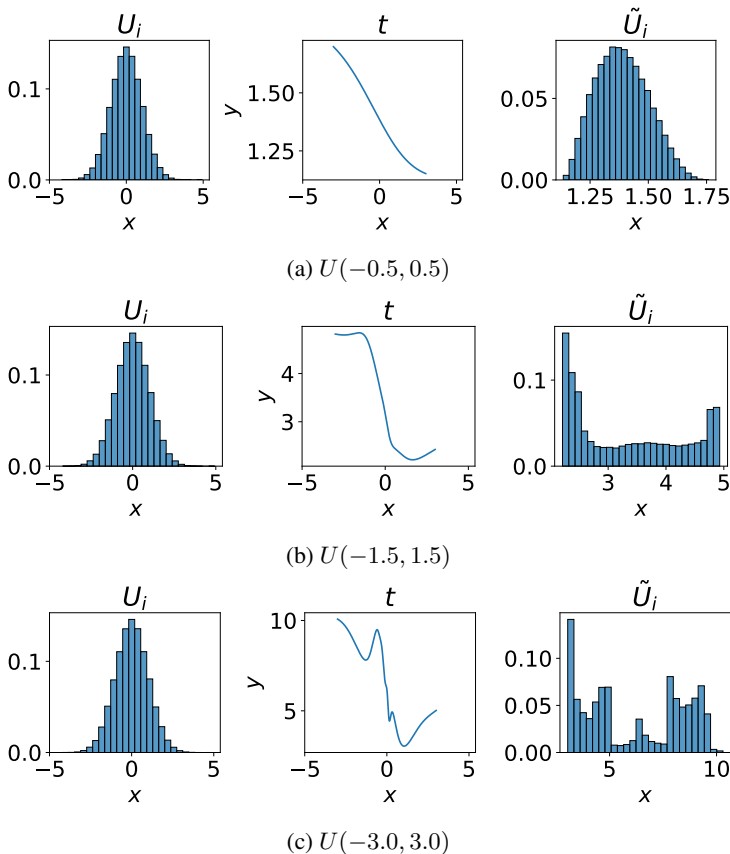

Figure 2: Gaussian noise (left) transformed via random nonlinear functions (center) to non-Gaussian *iid* noise (right). Weights of the MLP are sampled from either (a) $U(-0.5, 0.5)$, (b) $U(-1.5, 1.5)$, or (c) $U(-3.0, 3.0)$.

## B.4 Modeling of unfaithful distributions

Given a ground truth causal graph, we model an unfaithful distribution of the data by enforcing the cancellation of directed causal effects between pairs of nodes. In practice, we identify the fully connected triplets of nodes $X_i \rightarrow X_k \leftarrow X_j \leftarrow X_i$ in the ground truth, and we adjust the causal mechanisms such that the direct effect of $X_i$ on $X_k$ cancels out. In order to clarify the implementation details of our model, we consider a graph $\mathcal{G}$ with vertices $X_1, X_2, X_3$ and with the set of edges corresponding to the fully connected graph with trivial topological order $\pi = \{X_1, X_2, X_3\}$. We allow for mixed linear and nonlinear edges with liner effect on the nodes, such the set of structural equations is defined as:

$$
\begin{aligned}
X_1 &\coloneqq U_1, \\
X_2 &\coloneqq f(X_1) + U_2, \\
X_3 &\coloneqq f(X_1) - X_2 + U_3,
\end{aligned}
\tag{17}
$$

with $f$ nonlinear function. This definition of the mechanisms on $X_3$ cancels out $f(X_1)$ in the structural equation. In the case of large graphs with the number of nodes $d \in \{5, 10, 20, 50\}$ that we use in our experiments, we verify the unfaithful independencies in the data via kernel-based test of conditional independence [52], in correspondence of the pairs of nodes whose causal effect cancels out. We use a threshold of $0.05$ for the conditional independence testing.

## B.5 Dataset configurations

In this section, we extend the discussion of Section 3.1.2 which presents an overview of the parameters that define the different configurations for the generation of the synthetic datasets of our benchmark. We sample the ground truth structures according to four different algorithms for random graph

|          | 5 nodes     | 10 nodes | 20 nodes | 50 nodes |
|----------|-------------|----------|----------|----------|
| Sparse   | $p = 0.1^*$ | $m = 1$  | $m = 1$  | $m = 2$  |
| Dense    | $p = 0.4^*$ | $m = 2$  | $m = 4$  | $m = 8$  |

$^*$ Graphs are re-sampled such that they have at least 2 edges.

Table 2: Density schema for Erdos-Renyi graphs. The parameter $p$ denotes the probability of an edge between each pair of nodes in the graph, and $m$ denotes the average number of edges for each node in the graph. We scale the parameter $m$ with the number of nodes, such that the relative density (sparsity) is similar for all graph dimensions.

generation, and according to different specifications of density, number of nodes, and distribution of the noise terms. In the case of Erdos-Renyi (ER) generated graphs, we define the density of the edges relative to the number of nodes, according to the schema defined in Table 2. For the Scale-free (SF) model, we define the edge density in the graphs according to the same values of Table 2, but we do not generate SF graphs of 5 nodes. Similarly, we generate fully connected (FC) and Gaussian Random Partition (GRP) graphs only for $\{10, 20, 50\}$ nodes. FC generation does not require specifying any parameter for the density. In the case of GRP graphs, we use $p_{in} = 0.4$ as the probability of edges between a pair of nodes inside the same cluster, and $p_{out} = 0.1$ as the probability of edges between a pair of nodes belonging to different clusters.

For each of the graph configurations, we sample a ground truth and a dataset of observations generated according to one of the following scenarios (described in detail in Section 3.1.1):

- Vanilla additive noise model.
- PNL model, with invertible post nonlinear function $g(x) = x^3$ for each speficied structural equation.
- LiNGAM model, where the number of structural equations with linear mechanisms is parametrized by $\delta \in \{0.33, 0.66, 1.0\}$. The first two values of $\delta$ allow modeling mixed linear and nonlinear causal mechanisms, with respectively 33% and 66% of the structural equations being linear. Unless differently specified, we consider $\delta = 1$ when referring to the LiNGAM model.
- Confounded model, where the number of confounded pairs is parametrized by $\rho \in \{0.1, 0.2\}$, denoting the probability of two nodes having a common cause. Unless differently specified, we consider $\rho = 0.2$ when referring to the confounded model.
- Measurement error model, where the amount of variance explained by the additive error is parametrized by $\gamma \in \{0.2, 0.4, 0.6, 0.8\}$, denoting the inverse signal to noise ratio $\gamma := \frac{\text{Var}[\epsilon_i]}{\text{Var}[X_i]}$, with $X_i$ and $\epsilon_i$ defined in the structural equation (6). Unless differently specified, we consider $\gamma = 0.8$ when referring to the measurement error model.
- Unfaithful model, as discussed in the Appendix B.4.
- Autoregressive model, defined according to the structural equation (7) in order to simluate non-$iid$ samples in the data.

For each scenario, and for each parametrization that it admits, we generate a ground truth $\mathcal{G}$ according to each of the graph configurations specified at the beginning of the section, and a corresponding pair of datasets $\mathcal{D}$ of size 100 and 1000. The dataset generation is repeated under four possible distributions of the noise terms. In particular, each dataset has exogenous variables that are either normally distributed, or following a randomly generated non-Gaussian distribution, as discussed in Appendix B.2. Finally, in order to ensure statistically significant results, for each pair of graph and dataset configurations we generate $\mathcal{G}, \mathcal{D}$ according to 20 different random seeds.

## C   Benchmark methods

### C.1   CAM

CAM algorithm [46] infers a causal graph from data generated by an additive Gaussian noise model. First, it infers the topological ordering by finding the permutation of the graph nodes corresponding

to the fully connected graph that maximizes the log-likelihood of the data. After inference of the topological ordering, a pruning step is done by variable selection with regression. In particular, for each variable $X_j$ CAM fits a generalized additive model using as covariates all the predecessor of $X_j$ in the ordering, and performs hypothesis testing to select relevant parent variables. This is known as the *CAM-pruning* algorithm. For graphs with size strictly larger than 20 nodes, the authors of CAM propose an additional preliminary edge selection step, known as Preliminary Neighbours Search (PNS): given an order $\pi$, variable selection is performed by fitting for each $j = 1, \ldots, d$ an additive model of $X_j$ versus all the other variables $\{X_i : X_j \succ X_i \text{ in } \pi\}$, and choosing the $K$ most important predictor variables as possible parents of $X_j$. This preliminary search step allows scaling CAM pruning to graphs of large dimensions. In our experiments, CAM-pruning is implemented with the preliminary neighbours search only for graphs of size 50, with $K = 20$.

## C.2 RESIT

In RESIT (regression with subsequent independence test) [13] the authors exploit the independence of the noise terms under causal sufficiency to identify the topological order of the graph. For each variable $X_i$, they define the residuals $R_i = X_i - \mathbf{E}[X_i \mid \mathbf{X} \setminus \{X_i\}]$, such that for a leaf node $X_l$ it holds that $R_l = U_l - \mathbf{E}[U_l]$. The method is based on the property that under causal sufficiency, the noise variables are independent of all the preceding variables: after estimating the residuals from the data, it identifies a leaf in the graph by finding the residual $R_l$ that is unconditionally independent of any node $X_i, \forall i \neq l$ in the graph. Once an order is given, they select a subset of the edges admitted by the fully connected graph encoding of the ordering. We implement this final step with CAM-pruning.

## C.3 GraN-DAG

GraN-DAG [38] defines a continuous constrained optimization problem to infer the causal graph from an ANM with Gaussian noise terms. For each variable $X_i$ in the graph, the authors estimate the parameters of the conditional distribution $p(X_i \mid \mathbf{X} \setminus \{X_i\})$ with a neural network $\phi_i$. They define an adjacency matrix $A \in \mathbb{R}^{d \times d}$ representation of the causal DAG, by finding *inactive paths* in the neural network computations, where a path is defined as the sequence of weights of the network from the input $j$ to the output $k$: if zero weights are encountered in the path, then the output $k$ is independent of the input $j$. If this is repeated for all paths from $j$ to $k$ and for all outputs $k$, then all paths are inactive and $X_i$ is independent of $X_j$ conditional on the remaining variables, meaning that $A_{ij} = 0$. Note that GraN-DAG in principle does not require a post-processing consisting of an edge selection procedure, given that its output is not a fully connected encoding of a topological order, but an arbitrarily sparse graph. However, in practice, it is the case that GraN-DAG output approximates a fully connected graph, with a large number of false positives with respect to the ground truth edges (see Table 5 of Appendix A.3 in Lachapelle et al. [38] for quantitative results). To account for this, the authors of the method apply the CAM-pruning step on top of their neural network graph output. In our experiments, in order to compare the goodness of the ordering encoded by the GraN-DAG output before applying the CAM-pruning step, we sample one order at random between those admitted by the output adjacency matrix and compute its FNR-$\hat{\pi}$. Given that the order is selected at random, we consider an unbiased solution, as we show in Figure 3.

## C.4 DirectLiNGAM

ICA-LiNGAM [44] formulates a causal discovery algorithm for the identifiable LiNGAM model, assuming linear mechanisms and non-Gaussian noise terms. The idea is that solving for $\mathbf{X}$ the system defined in Equation (3), one obtains

$$\mathbf{X} = \mathbf{A}\mathbf{U}, \tag{18}$$

where $\mathbf{A} = (\mathbf{I} - \mathbf{B})^{-1}$. By standard linear ICA (independent component analysis) it is possible to find $\mathbf{A}$, which is equivalent to finding the weighted adjacency matrix $\mathbf{B}$. This intuition lies at the base of the DirectLiNGAM algorithm [44], a variation of ICA-LiNGAM that uses pairwise independence measures to find the topological order of the graph, and covariance-based regression to find the connection strengths in the matrix $\mathbf{B}$.

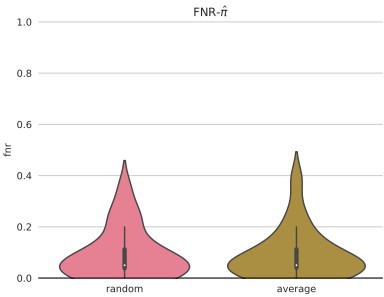

Figure 3: In order to evaluate the goodness of the inferred ordering of GraN-DAG, we sample one topological order at random between those admitted by the adjacency matrix before the CAM-pruning step. In this figure, we compare the empirical FNR-$\hat{\pi}$ of an order randomly sampled between those admitted by the output, against the average of the FNR-$\hat{\pi}$ computed on the set of all possible orderings admitted by the output. We see that selecting an order at random gives an unbiased representation of the average order accuracy, between those admitted by GraN-DAG output before the CAM-pruning. The violin plots refer to the FNR-$\hat{\pi}$ evaluated on ER graphs with 10 nodes over 20 different random seeds.

## C.5 PC

PC algorithm (Section 5 of Spirtes et al. [4]) is a causal discovery method based on conditional independence testing that finds a CPDAG from the data. First, it starts from a fully connected undirected graph, and estimates the skeleton of the graph by removing edges between each pair of nodes $X_i, X_j$ if it finds a subset $\mathbf{S} \subset \mathbf{X} \setminus \{X_i, X_j\}$ such that $X_i \perp\!\!\!\perp X_j \mid \mathbf{S}$. Then, it finds all the v-structures $X_i \to X_j \leftarrow X_k$ along with their directions. Finally, additional orientation rules are applied to direct as many edges as possible in the output CPDAG. In our experiments, we use a kernel-based test of conditional independence [52].

## C.6 GES

The GES algorithm [9] (Greedy Equivalent Search) defines a discrete optimization problem over the space of all CPDAGs, and outputs the graph that maximizes the fit measure according to some score (e.g. the Bayesian Information Criterion (BIC) score). The algorithm is defined as a two steps greedy procedure. It starts from an empty graph, and in the *forward* step it adds directed edges one by one, each time selecting the directed edge that most increases the fit score. When edge addition doesn't improve the score any further, in the *backward* step it removes edges one by one until the score stops increasing. The DAG defined by this procedure is then transformed into a CPDAG, such that GES final output is a Markov equivalence class.

## C.7 SCORE

Rolland et al. [23] defines a formal criterion for the identification of the causal order of a graph underlying an additive noise models with Gaussian distribution of the noise terms $U_i \sim \mathcal{N}(0, \sigma_i)$. Under these assumptions, the score entry of a leaf node $X_l$ is $s_l(\mathbf{X}) = -\frac{X_l - f_l(\mathrm{PA}_l)}{\sigma_l^2}$. It is then easy to verify that $\partial_{X_l} s_l(\mathbf{X}) = -\frac{1}{\sigma_j^2}$, such that the diagonal entry of the score's Jacobian associated to a leaf node is a constant. Based on this relation, a formal criterion identifying leaf nodes holds:

**Lemma 1** (Lemma 1 of [23]). *Let $\mathbf{X}$ be a random vector generated according to an identifiable ANM with exogenous noise terms $U_i \sim \mathcal{N}(0, \sigma_i^2)$, and let $X_i \in \mathbf{X}$. Then*

$$\mathrm{Var}\left[\partial_{x_i} s_i(\mathbf{X})\right] = 0 \iff X_i \text{ is a leaf}, \ \forall i = 1, \ldots, d. \tag{19}$$

The Lemma is exploited by SCORE algorithm for estimation of the topological order, given a dataset of i.i.d. observations $X \in \mathbb{R}^{n \times d}$: first it estimates the diagonal elements of the Jacobian matrix of the score $J(s(\mathbf{X}))$ via score matching [22]. Then, it identifies a leaf in the graph as the $\arg\min_i \mathrm{Var}[\partial_{x_i} s(\mathbf{X})]$, which is removed from the graph and assigned a position in the order vector. By iteratively repeating this two steps procedure up to the source nodes, all variables in $\mathbf{X}$ end up

being assigned a position in the causal ordering. Finally, SCORE applies the CAM-pruning algorithm to select a subset of the edges in the fully connected DAG encoding of the inferred topological order.

## C.8 NoGAM

Montagna et al. [25] proposes a generalization of SCORE, defining a formal criterion for the identification of leaf nodes in a graph induced by an additive noise model without restrictions on the distribution of the noise terms. After some manipulations, it can be shown that the score entry of a leaf $X_l$ defined in Equation (9) satisfies

$$s_l(\mathbf{X}) = \partial_{U_l} \log p_l(U_l), \tag{20}$$

such that observations of the pair $(U_l, s_l(\mathbf{X}))$ can be used to learn a predictor of the score entry. For an additive noise model, the authors show that the noise term of a leaf is equal to the residual defined as:

$$R_l := X_l - \mathbf{E}\left[X_l \mid \mathbf{X} \setminus X_l\right]. \tag{21}$$

Then, it is possible to find a consistent approximator of the score entry of a leaf node using $R_l$ as the only predictor.

**Lemma 2** (Lemma 1 of [25]). *Let $\mathbf{X}$ be a random vector generated according to an identifiable ANM, and let $X_i \in \mathbf{X}$. Then*

$$\mathbf{E}\left[\left(\mathbf{E}\left[s_i(\mathbf{X}) \mid R_i\right] - s_i(\mathbf{X})\right)^2\right] = 0 \iff X_i \text{ is a leaf.}$$

Similarly to SCORE, NoGAM algorithm defines a procedure for estimation of the topological order by iterative identification of leaf nodes, which are found as the $\operatorname{argmin}_i \mathbf{E}\left[\left(\mathbf{E}\left[s_i(\mathbf{X}) \mid R_i\right] - s_i(\mathbf{X})\right)^2\right]$. In practice, the residuals $R_i, i = 1, \ldots, d$, can be estimated by any regression algorithm, whereas the score is approximated by score matching with Stein identity [22].

## C.9 DAS

Montagna et al. [24] defines a condition on the Jacobian of the score function that identifies the edges of the graph induced by an additive noise model with Gaussian distribution of the noise terms, given a valid causal order.

**Lemma 3** (Lemma 1 of [24]). *Let $\mathbf{X}$ be a random vector generated according to an identifiable ANM with exogenous noise terms $U_i \sim \mathcal{N}(0, \sigma_i^2)$, and let $X_l \in \mathbf{X}$ be a leaf node. Then:*

$$\mathbf{E}\left[\left|\partial_{x_j} s_l(\mathbf{X})\right|\right] \neq 0 \iff X_j \in \mathrm{PA}_l(\mathbf{X}), \ \forall j \in \{1, \ldots, d\} \setminus \{l\} \tag{22}$$

In practice, off-diagonal elements of the Jacobian matrix contain information about conditional independencies of the variables in the DAG, such that they define a condition of identification of the graph edges. Given the ordering procedure of SCORE, DAS (acronym for Discovery At Scale) defines an algorithm that can map the score function to a unique causal graph with directed edges: in practice, condition (22) of Lemma 3 is verified via hypothesis testing for the mean equals to zero. Note that, despite the fact that this approach provides a consistent estimator of the causal graph, the authors of DAS retain a CAM-pruning step on top of their edge selection procedure based on Lemma 3, in order to reduce the number of false positives of the inferred output. The benefit of DAS preliminary edge detection is that it reduces the computational costs of CAM-pruning, which is cubic in the number of nodes in the graph, such that it doesn't scale well to high dimensional graphs. Overall, given an input dataset $X \in \mathbb{R}^{n \times d}$, with $n$ number of samples and $d$ number of nodes in the graph, DAS computational complexity is $\mathcal{O}(dn^3 + d^2)$, whereas, for the SCORE algorithm this is $\mathcal{O}(dn^3 + nd^3)$.

## C.10 Random Baseline

In our experimental analysis of Section 4, we consider the performance of a random baseline in terms of F1 score and FNR-$\hat{\pi}$ accuracy of the order (Figure 1). Our random baseline is defined as follows. Given a graph with $d$ variables, we sample a random topological order $\pi$ as a permutation of the vector of elements $X_1, \ldots, X_d$. Then, given the fully connected graph admitted by the order, with the set of edges $\mathcal{E}_\pi = \{X_{\pi_i} \to X_{\pi_j} : X_{\pi_i} \prec X_{\pi_j}, \forall i, j = 1, \ldots, d\}$, for each pair of connected nodes we sample a Bernoulli random variable $Y$ with parameter $p = 0.5$, such that the edge is removed for $Y = 0$.

# D   Metrics definition

For the evaluation of the experimental results of our benchmark, we consider the F1 score, the false positive (FP) and false negative (FN) rates of the inferred graph, and the false negative rate FNR-$\hat{\pi}$ of the fully connected encoding of the output topological order. In order to specify the F1 score, we need a definition of FP, FN, and true positive (TP), that applies to both undirected and directed edges, given that we evaluate both DAGs and CPDAGs.

- We define as TP any predicted edge that is in the skeleton of the ground truth graph (i.e. the set of edges that doesn't take direction into account).
- We define the FPs as the edges in the skeleton of the predicted graph that are not in the skeleton of the ground truth graph. Note that this definition of FP doesn't penalize undirected edges or edges inferred with reversed direction.
- We define as FN a pair of nodes that are disconnected in the predicted skeleton while being connected in the ground truth. Additionally, we count as false negatives inferred edges whose direction is reversed with respect to the DAG ground truth.

Then, the F1 score is defined as the ratio $\frac{TP}{TP+0.5(FN+FP)}$.

# E   Possible generalisation of NoGAM to the PNL model

Proposition 1 of Section 3.2.1 suggests that it is possible to generalize Lemma 2 and, accordingly, the NoGAM algorithm, to the case of the post nonlinear model.

## E.1   Proof of Proposition 1

*Proof.* Let $\mathbf{X} \in \mathbb{R}^d$ be a random vector generated by the post nonlinear model of Equation (5). Given the Markov factorization of Equation (2), the logarithm of the joint distribution $p_{\mathbf{X}}$ satisfies the following equation:

$$\log p_{\mathbf{X}}(\mathbf{X}) = \sum_{i=1}^{d} \log p_{X_i}(X_i).$$

Then, for a node $X_i$ in the graph the score entry $s_i$ is defined according to Equation (8), whereas given a leaf node $X_l$ in the graph, $s_l$ satisfies the following:

$$s_l(\mathbf{X}) := \partial_{x_l} \log p_{\mathbf{X}}(\mathbf{X}) = \partial_{x_l} \log p_l(X_l|\mathrm{PA}_l). \tag{23}$$

Our goal is to show that $\partial_{x_l} \log p_l(X_l|\mathrm{PA}_l) = \partial_{x_l} \log p_l(U_l)$, with $U_l = g^{-1}(X_l) - f_l(\mathrm{PA}_l)$ and $g^{-1}$ the inverse of the postnonlinear function $g$ (which is invertible by modeling assumption). As a notational remark, in what follows we will drop any sub-index on the distribution of the random variables, which we distinguish by their argument. Also, we denote *realizations* of random variables (or random vectors) with lowercase letters (e.g. $x_l$ is the value of the random variable $X_l$). We rewrite the distribution of $X_l$ conditional on its parents by marginalizing over all values of $U_l$:

$$p(x_l \mid \mathrm{pa}_l) = \int_{u_l} p(x_l \mid \mathrm{pa}_l, u_l) p(u_l) du_l \tag{24}$$

$$= \int_{u_l} p(x_l \mid \mathrm{pa}_l, u_l) p(u_l) \mathbb{1}(x_l = g(f_l(\mathrm{pa}_l) + u_l)) du_l \tag{25}$$

$$= \int_{u_l} p(x_l \mid \mathrm{pa}_l, u_l) p(u_l) \mathbb{1}(u_l = g^{-1}(x_l) - f_l(\mathrm{pa}_l)) du_l, \tag{26}$$

with $\mathbb{1}$ being the indicator function. Being $g$ an invertible function, the value of $u_l$ equals to $g^{-1}(x_l) - f_l(\mathrm{pa}_l)$ is unique, which implies that $p(x_l \mid \mathrm{pa}_l, u_l) = 0$ if $u_l \neq g^{-1}(x_l) - f_l(\mathrm{pa}_l)$, else $p(x_l \mid \mathrm{pa}_l, u_l) = 1$. Let us denote $u_l^* := g^{-1}(x_l) - f_l(\mathrm{pa}_l)$. Then, the integral in Equation 26 simply becomes:

$$p(x_l \mid \mathrm{pa}_l) = \int_{u_l} dp(u_l) \mathbb{1}(u_l = u_l^*) = p(u_l^*). \tag{27}$$

Thus, $\partial_{x_l} \log p(X_l|\mathrm{PA}_l) = \partial_{x_l} \log p(U_l)$. $\qquad\square$

### E.2 Discussion

Proposition 1 derives a connection between Lemma 2 defined by Montagna et al. [25] for identifiable additive noise models to the case of a PNL model. Note that the authors define a consistent estimator of $s_l$ score function of a leaf node $X_l$ from the residual $R_l := X_l - \mathbf{E}\left[X_l \mid \mathbf{X} \setminus X_l\right]$, which satisfies $R_l = U_l$ in the case of an ANM with noise terms centered at zero. In general, the latter equality does not hold for a post nonlinear model, meaning that regression of a leaf variable against all the other variables of $\mathbf{X}$ does not guarantee a consistent estimation of the disturbance on the leaf structural equation. This implies that, as is, NoGAM doesn't provide theoretical guarantees of consistent estimation of the topological order of a PNL model.

## F Proof of Proposition 2

We define two lemmas preliminary to the proof of Proposition 2.

**Lemma 4.** *Let $\mathbf{X} \in \mathbb{R}^d$ be generated according to an SCM $\mathcal{M}$ that satisfies score-sortability, and let $\mathcal{G}$ be the graph induced by the model. Then, there exists a leaf node of $\mathcal{G}$ that is score-identifiable.*

*Proof.* By contradiction, let's say that the node $X_l$ with $l = \operatorname{argmin}_i \operatorname{Var}[s_i(\mathbf{X})]$ is not a leaf node. Then, the causal order $\pi$ where $X_l$ is a successor of all other nodes in the graph, is not a correct ordering, implying that the model is not *score-sortable*. $\square$

**Lemma 5.** *Let $\mathbf{X} \in \mathbb{R}^d$ be generated according to an SCM $\mathcal{M}$ that satisfies score-sortability, and let $\mathcal{G}$ be the graph induced by the model. Let $\mathcal{M}_{\setminus\{l\}}$ the model defined removing the leaf node $X_l$ from the set of structural equations of $\mathcal{M}$. Then, the model $\mathcal{M}_{\setminus\{l\}}$ is score-sortable.*

*Proof.* By contradiction, let's assume that $\mathcal{M}_{\setminus\{l\}}$ does not satisfy *score-sortability*, such that the node $X_m$ with $m = \operatorname{argmin}_{i=1,\ldots,l-1,l+1,\ldots,d} \operatorname{Var}[s_i(\mathbf{X})]$ is not a leaf node in the graph $\mathcal{G}_{\setminus\{l\}}$ induced by $\mathcal{M}_{\setminus\{l\}}$. Then, any topological order $\pi$ with $X_m$ successor of all nodes $X_i, i = 1, \ldots, l-1, l+1, \ldots, d$, is a wrong topological ordering of the graph $\mathcal{G}$. This implies that $\mathcal{M}$ is not *score-sortable*. $\square$

Now, we present the proof of Proposition 2.

*Proof.* (Proof of Proposition 2.) By Lemma 4, being $\mathcal{M}$ a *score-sortable* model, there exists a leaf $X_l$ such that $l := \operatorname{argmin}_i \operatorname{Var}[s_i(\mathbf{X})]$. Then,

$$\operatorname{Var}[\partial_l \log p_l(X_l \mid \operatorname{PA}_l)] \leq \operatorname{Var}[\partial_i \log p_i(X_i \mid \operatorname{PA}_i)] + \sum_{k \in \operatorname{CH}_i} \operatorname{Var}[\partial_k \log p_k(X_k \mid \operatorname{PA}_k)] + C,$$

for all $i = 1, \ldots, d$. Moreover, by previous Lemma 5, the model $\mathcal{M}_{\setminus\{l\}}$ is *score-sortable*. Thus there exists an index $m \in \{1, \ldots, l-1, l+1, \ldots, d\}$ such that $X_m$ is a leaf and $\operatorname{Var}[s_m(\mathbf{X}_{\setminus\{l\}})] \leq \operatorname{Var}[s_i(\mathbf{X}_{\setminus\{l\}})], \forall i = 1, \ldots, l-1, l+1, \ldots, d$. Then, the topological ordering defined by iterative identification of leaf nodes with Lemma 4 and Lemma 5 on the subgraphs resulting by removal of a leaf node, is correct with respect to the model $\mathcal{M}$. $\square$

## G Tuning of the hyperparameters in the experiments

The methods included in the benchmark require the tuning of several hyperparameters for the inference procedure. In particular, PC, DAS, SCORE, NoGAM, RESIT, GraN-DAG, CAM, and DiffAN require a threshold $\alpha$ over the p-value of the statistical test used for the edge selection procedure. Instead, GES applies a regularization term weighted by $\lambda$ to its score, which penalizes the number of edges included in the inferred graph: the higher the value of $\lambda$, the sparser the solution. Given that the tuning of both $\alpha$ and $\lambda$ requires prior knowledge about the sparsity of the ground truth, there is no established procedure for finding their optimal values in real-world settings, where the ground truth is not accessible. Thus, in order to enable a fair comparison between all the methods, we always select the optimal value of $\alpha \in \{0.001, 0.01, 0.05, 0.1\}$ and $\lambda \in \{0.05, 0.5, 2, 5\}$ over each benchmark dataset. In Section H we discuss the stability of the algorithms with respect to choices of these hyperparameters.

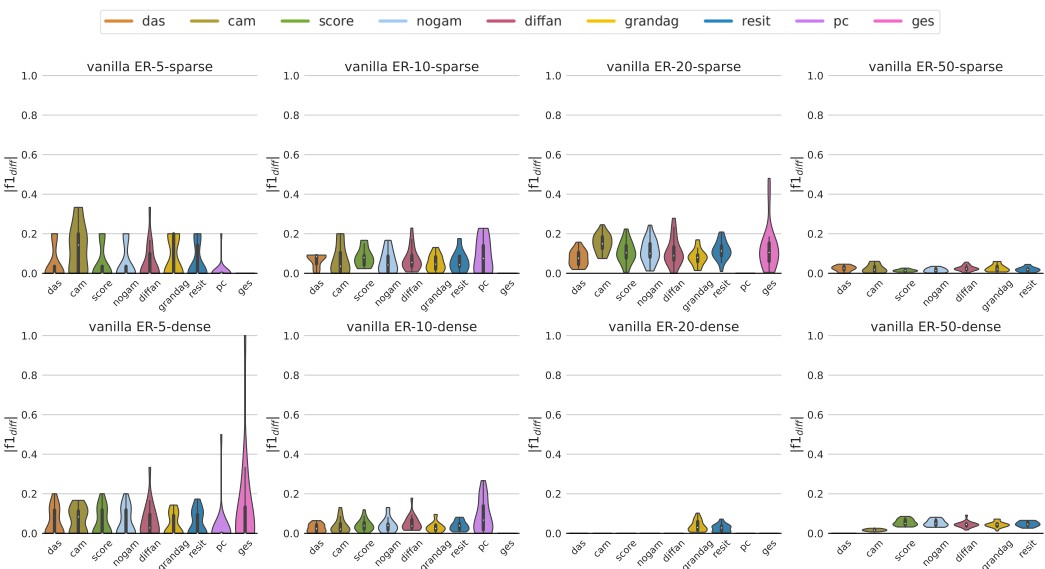

Figure 4: The violin plots in the figure represent the difference between the F1 score of a method running inference with hyperparameters optimized using the ground truth, versus the F1 score of the same method using a default value of the hyperparameters. We denote this difference with $|\text{f1}_{\text{diff}}|$. In the case of GES, we define as default $\lambda = 0.5$. For all the remaining methods, the default alpha threshold is defined as $\alpha = 0.05$. The violin plots refer to the inference performance on datasets and graphs generated according to 20 different random seeds. Results in the table are on data generated from the vanilla scenario, and we consider Erdos-Renyi graphs with the number of nodes in $\{5, 10, 20, 50\}$ in the dense and sparse settings.

GraN-DAG and DiffAN both define a learning procedure over the data, which requires the tuning of several training hyperparameters, the most important of which is the learning rate $\eta$. For each dataset, this is optimized over the loss function on a held-out validation set, without accessing the ground truth graph.

## H  Stability with respect to hyperparameters choices

Most causal discovery methods come with hyperparameters that alleviate minor assumption violations (e.g. sparsity regularization or higher thresholds on p-values in statistical tests). In the absence of background knowledge, tuning these hyperparameters is an art that often relies on pre-conceptions about reasonable solutions. In this section, we investigate the impact of these hyperparameters on the accuracy of the output graph. GES penalizes dense solution with a regularization term in its score, weighted by a hyperparameter $\lambda$ that can not be tuned in the absence of the ground truth. Similarly, an $\alpha$ threshold on p-values for statistical tests for edge selection is required by all the benchmarked methods (excluding GES and DirectLiNGAM) and can not be tuned without knowledge of the ground truth. In this section, we analyze the inference F1 score by fixing $\alpha$ and $\lambda$ to the commonly accepted default values of $0.05$ and $0.5$ respectively. In Figure 4 we summarise the absolute value of the difference between the F1 score obtained with hyperparameters optimized on the ground truth of each dataset, against the F1 score yielded by inference with the default $\alpha$ and $\lambda$ values (denoted with $|\text{f1}_{\text{diff}}|$ in the plots). According to our empirical findings, in the case of graphs with at least 10 nodes, the median of this difference is in general lower than $0.1$, and most of the time close to $0$. Sparse graphs seem to be more affected in their performance by the hyperparameters choice: this means that using the default $\alpha$ and $\lambda$ causes an increase of false positives in the output graph. Bühlmann et al. [46] shows that under correct topological order, a graph whose set of edges is a superset of the ground truth still provides consistent estimates of the causal effects, such that increasing the false positives doesn't affect the outcome of downstream tasks, but only the statistical efficiency of the inference. Given that estimation of the topological ordering is not affected by the choice of $\alpha$ and $\lambda$ values, we suggest that the role of hyperparameters value is in this respect marginal with respect to the task of interest.

**Implications.** We observe remarkable stability of the benchmarked methods with respect to the choice of their hyperparameters. The biggest drops in F1 score are observed on sparse graphs, meaning that the default parameters cause an increase of false positives, which nevertheless does not affect the downstream task of interest of consistent estimation of causal effects.

# I    Other experimental results on Erdos-Renyi graphs

In this section, we present additional experimental results on Erdos-Renyi graphs.

## I.1    The effect of non-*iid* distribution of the data

Figure 1f (right) illustrates that all the methods included in our benchmark do not perform well on data sampled from a non-*iid* distribution generated according to the autoregressive model of Equation (7): F1 score and FNR-$\hat{\pi}$ are indeed similar to that of the random baseline. It clearly appears that none of the presented algorithms provide guarantees of good empirical performance in the setting of non-*iid* distribution of the data.

## I.2    Experiments under arbitrary distribution of the noise terms

In Section 2.2 we discussed the effect of the distribution of the noise terms on the identifiability of the causal graph underlying an SCM. Given that the assumption of Gaussian distribution of the disturbances is often not satisfied in real datasets, it is important to provide empirical evidence on the performance of the benchmarked methods on data generated with an arbitrary distribution of the noise. In this section, we discuss experiments on data generated with the noise terms that are *iid* samples from the distribution of Figure 2c. Similar to Section 4, we analyze results on ER graphs with 20 nodes, with experiments repeated over 20 random seeds. In this section, we include results of DirectLiNGAM, on both linear and nonlinear SCMs.

Figure 5 illustrates the FNR-$\pi$ score of the inferred topological order on data generated according to the vanilla model with non-Gaussian noise terms. Under these conditions, NoGAM and RESIT provide theoretical guarantees of consistent estimate of the causal ordering. Similarly, PC and GES do not make explicit assumptions on the distribution of the noise terms (despite the fact that GES optimizes a Gaussian likelihood). SCORE, DiFFAN, DAS, and CAM instead are limited by restrictions on the noise terms, which are required to be normally distributed. However, Figure 5 (right) shows that, except for CAM, they can estimate the order with accuracy comparable to that achieved in the case of *vanilla* generated data, with Gaussian distribution of the disturbances. These observations are in line with the experimental findings in Montagna et al. [25], which shows how the structure of the score entries of leaf nodes can still be exploited by SCORE for inference on data generated under arbitrary noise distribution. Our experimental results agree with this intuition: surprisingly, SCORE ordering ensures better FNR-$\hat{\pi}$ accuracy than RESIT, despite the latter being explicitly designed to be insensitive to the distribution of the noise terms. Interestingly, we notice that the median of the violin plot referred to DirectLiNGAM in Figure 5 (right) is close to that of RESIT and CAM: this suggests that in the realistic scenario of mixed linear and nonlinear mechanisms with non-Gaussian additive disturbances, we can expect DirectLiNGAM to give performance significantly better than several methods designed to perform on nonlinear ANM. Figure 5 (left), shows that the in the case of methods whose ordering accuracy is comparable to the Gaussian case, the F1 score after pruning is also comparable to that on Gaussian data. This means that CAM-pruning is robust with respect to arbitrary distributions of the noise terms. Additional experimental results on data generated according to the misspecified scenarios of Section 3.1.1 with non-Gaussian distribution of the disturbances, are presented in Figure 6.

**Implications.** Most of the benchmarked methods are capable of robust inference on datasets generated by an ANM with non-Gaussian noise terms. DirectLiNGAM shows remarkable performance, comparable to that of several methods designed for inference on nonlinear additive noise models.

## I.3    Experiments with score-sortability

In this section, we present the experimental results of a simple ordering algorithm, that we name ScoreSort, based on the *score-sortability* criterion defined in Section 4.1.2. Given the random vector

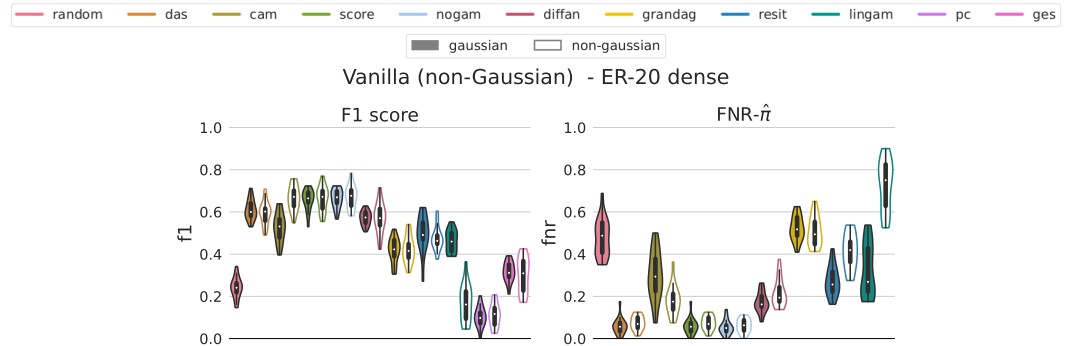

Figure 5: F1 score and FNR-$\hat{\pi}$ on data generated with non-Gaussian distribution of the noise terms (c.f. Figure 2c). For each method, we also display the violin plot of its performance on the *vanilla* scenario with Gaussian noise terms, with transparent color. F1 score (the higher the better) and FNR-$\hat{\pi}$ (the lower the better) are evaluated over 20 seeds on Erdos-Renyi dense graphs with 20 nodes (ER-20 dense). FNR-$\hat{\pi}$ is not computed for GES and PC methods, whose output is a CPDAG.

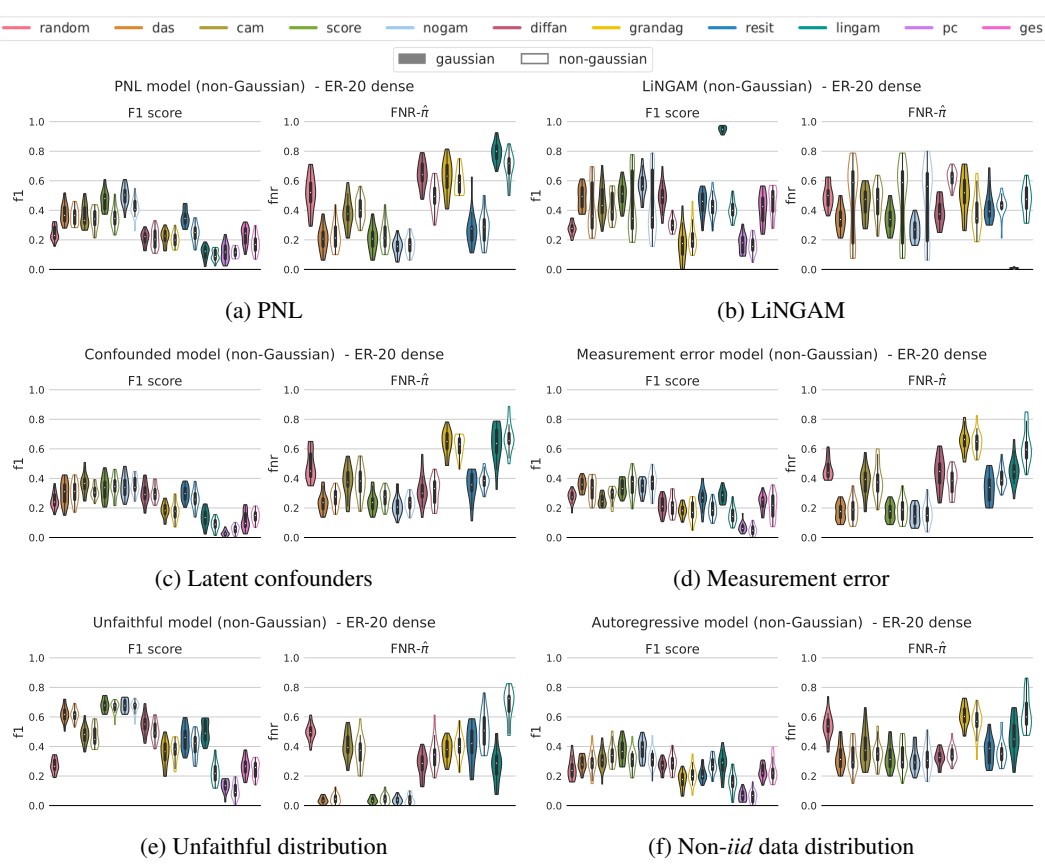

Figure 6: Experimental results on the misspecified scenarios with non-Gaussian distribution of the noise terms. For each method, we also display the violin plot of its performance on the same misspecified scenario under Gaussian distribution of the noise terms, with transparent color. F1 score (the higher the better) and FNR-$\hat{\pi}$ (the lower the better) are evaluated over 20 seeds on Erdos-Renyi dense graphs with 20 nodes (ER-20 dense). FNR-$\hat{\pi}$ is not computed for GES and PC, methods whose output is a CPDAG.

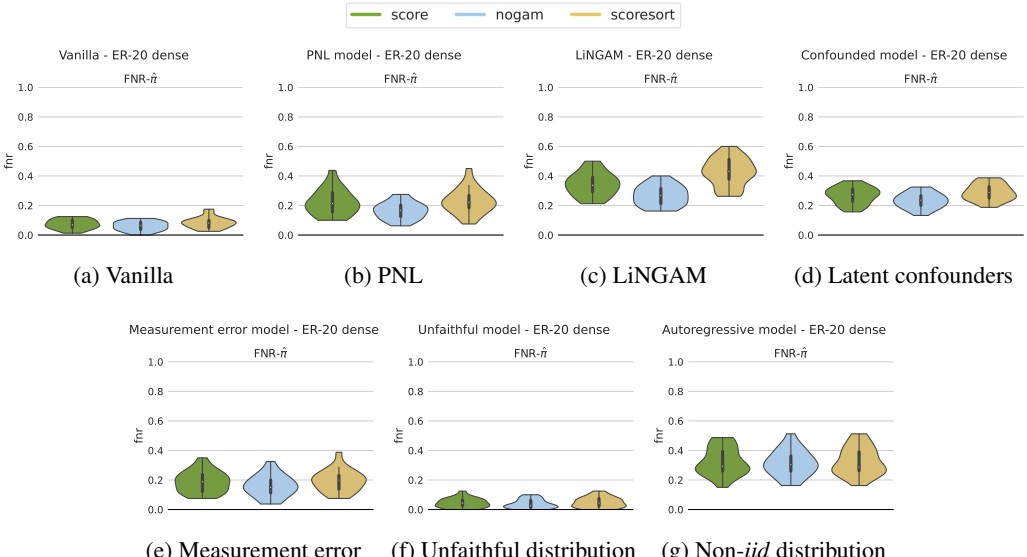

Figure 7: Experiments on *score-sortability*. We compare the FNR-$\hat{\pi}$ accuracy of the simple ScoreSort baseline (c.f. Algorithm 1) with SCORE and NoGAM algorithms performance. The violin plots are evaluated over 20 seeds on Erdos-Renyi dense graphs with 20 nodes (ER-20 dense).

$\mathbf{X} \in \mathbb{R}^d$ generated according to a structural causal model, the ScoreSort baseline identifies the index of a leaf node $l$ as the $\operatorname{argmin}_i \operatorname{Var}[s_i(\mathbf{X})]$. Then it removes the leaf node $X_l$ from the set of vertices of the graph, and identifies the next leaf with the $\operatorname{argmin}$ of the variance of the score vector of the remaining set of nodes. Identification of leaf nodes according to this procedure over the $d$ (sub)graphs obtained by the iterative leaves removal yields a topological ordering $\pi$ for the graph $\mathcal{G}$ underlying the SCM. If the model is *score-sortable*, then, according to Proposition 2, the causal order $\pi$ is correct with respect to the graph. Details on ScoreSort are presented in the Algorithm box 1. In practice, ScoreSort estimates the score vector $\hat{s}$ according to the same score-matching algorithm used by SCORE and NoGAM, which is based on the Stein identity [22].

Figure 7 compares ScoreSort performance with NoGAM and SCORE ordering algorithms, on data generated according to the vanilla and misspecified scenarios of Section 3.1.1. In line with our considerations in the discussion on score matching robustness in Section 4.1.2, we observe that ScoreSort FNR-$\hat{\pi}$ accuracy is comparable, with statistical significance, to that of SCORE and NoGAM. This is true both in the case of data generated under vanilla and misspecified scenarios.

---

**Algorithm 1** ScoreSort algorithm for inference of the causal order

---

Input: data matrix $X \in \mathbb{R}^{n \times d}$
$\pi \leftarrow [\,]$
$nodes \leftarrow [1, \ldots, d]$
**for** $i = 1, \ldots, d$ **do**
    $\hat{\mathbf{s}} \leftarrow \texttt{score-matching}(X)$
    $l_{index} \leftarrow \operatorname{argmin} \hat{\operatorname{Var}}[\hat{\mathbf{s}}]$
    $l \leftarrow nodes[l_{index}]$
    $\pi \leftarrow [l, \pi]$
    Remove $l_{index}$-th column from $X$; Remove $l$ from $nodes$
**end for**
$\pi \leftarrow reverse(\pi)$ (first node is a source, last node is a leaf)
**return** $\pi$

---

# J   Experiments on SF, GRP, and FC graphs

In this section, we analyze the F1 score and FNR-$\hat{\pi}$ accuracy of the benchmarked methods in the case when the ground truth graph is generated with Scale-free, Fully Connected, and Gaussian Random Partitions algorithms.

## J.1   Experiments on Scale-free graphs

Figure 8 illustrates the F1 score and FNR-$\hat{\pi}$ on SF graphs. We see that similar to the case of ER networks, score matching-based methods show remarkable robustness in the inferred order in the case of several misspecified scenarios, particularly, on data generated by the PNL (Figure 8a right), measurement error (Figure 8d right), and unfaithful models (Figure 8e right). However, we notice two significant differences with respect to the conclusions that we derived in the case of ER graphs in Section 4, Figure 1: in the case of the LiNGAM model, SCORE, DAS and NoGAM display FNR-$\hat{\pi}$ accuracy that is remarkably close to that on vanilla data (Figure 8b right), whereas their decrease in performance in the case of latent confounders effects (Figure 8c right), is worse than that observed on ER graphs. Interestingly, the results on the F1 score show that DAS, SCORE, NoGAM, and DiffAN performance is surprisingly good (with respect to the random baseline) across all the misspecified scenarios, which suggests good performance of CAM-pruning on SF graphs. Moreover, we see that GraN-DAG and RESIT inference procedure is close to that of the random baseline in almost all the misspecified scenarios: this is also explained by the poor performance of these two methods on vanilla data and SF graphs (illustrated in the transparent violin plots of Figure 8).

**Implications.** Score-matching based approaches show remarkable robustness even in the case of SF graphs. Interestingly, CAM-pruning performance on SF graphs is generally better than the one relative to ER-generated ground truths, such that the observed F1 score is often better than random. We also observe that RESIT and GraN-DAG ordering ability is negatively affected by the SF ground truth, in comparison to the case of ER graphs.

## J.2   Experiments on fully connected graphs

In the case of fully connected graphs, the ground truth admits a unique topological ordering. This means that we expect to observe an increase in the false negative rate FNR-$\hat{\pi}$, with respect to the results on ER graphs of Figure 1. This is in line with our empirical evidence, as illustrated in Figure 9. However, we see that score matching-based approaches still show robust performance in the inference of the ordering with respect to misspecified scenarios, except for the case of data generated according to the LiNGAM ground truth model. Notably, the F1 score accuracy of GES is consistently better than that of all the other methods, across every scenario. This is to be understood with the fact that the unpenalized BIC score optimized by GES always improves by increasing the number of edges in the graph. Given that we optimize the regularizer term $\lambda$ on each dataset, the optimal $\lambda$ value will naturally privilege the densest solutions. Different is the case for methods that rely on CAM-pruning, which display an F1 score consistently lower than the random baseline, except for the case of data generated by the unfaithful and LiNGAM models.

**Implications.** Score matching-based approaches are in general robust to misspecifications of the scenario in the case of a fully connected ground truth. GES shows a remarkable performance, that is partly explained by the optimization of the loss penalization term directly on the ground truth. Finally, we observe that the CAM-pruning step is negatively affected by the large density of the graphs.

## J.3   Experiments on GRP graphs

In Figure 10 we see that score matching-based methods and CAM algorithm display better robustness in the inference of the order than the remaining approaches, in reference to all of the benchmarked scenarios. The FNR-$\hat{\pi}$ of RESIT, GraN-DAG, and DiffAN are significantly close to the random baseline for data generated according to most of the ground truth models (with the exception of DiffAN on the LinGAM model and GraN-DAG on unfaithful samples). In terms of F1 score, most of the methods show good capability of inferring the ground truth graph, even in the case of data generated under assumption violations. Note that the F1 score of the random ground truth is remarkably bad, if compared to the case of SF, FC, and ER graphs. This is in line with the cluster structure of GRP graphs: given that the random baseline connects pair of nodes all with the same

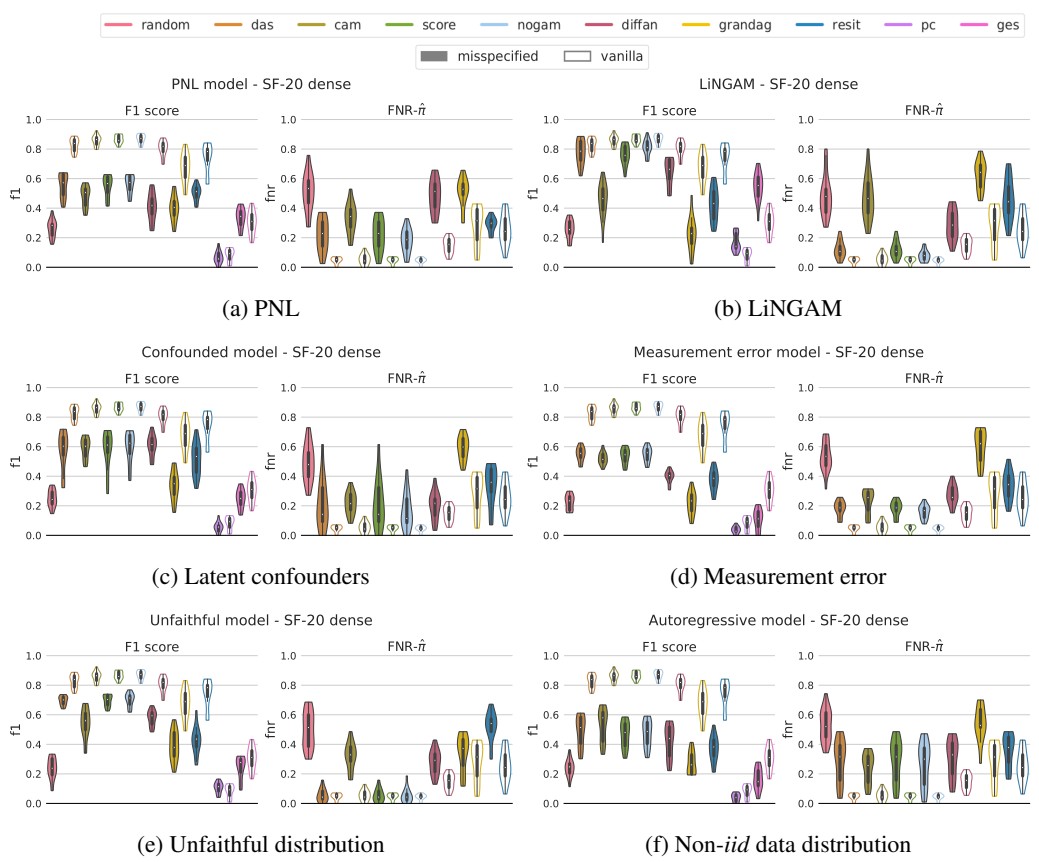

Figure 8: Experimental results on the misspecified scenarios. For each method, we also display the violin plot of its performance on the *vanilla* scenario with transparent color. F1 score (the higher the better) and FNR-$\hat{\pi}$ (the lower the better) are evaluated over 20 seeds on Scale-free dense graphs with 20 nodes (SF-20 dense). FNR-$\hat{\pi}$ is not computed for GES and PC methods, whose output is a CPDAG. Note that DirectLiNGAM performance does not appear, as both the linear mechanisms and non-Gaussian noise assumptions are violated.

probability 0.5, we expect a large number of false positives due to edges between nodes of different clusters.

**Implications.** Score matching-based approaches and CAM algorithm are remarkably robust to model misspecification both in terms of F1 score and FNR-$\hat{\pi}$ accuracy.

## K   Other results

**Statistical efficiency.** Figure 11 shows F1 score and FNR-$\hat{\pi}$ accuracy on datasets with sample size equals to 100 and 1000. Comparing the relative difference in performance with respect to different sample sizes, we get an empirical idea of the statistical efficiency of the inference methods. In line with our expectations, the experimental results show that both metrics are negatively affected by the reduction in sample size. Interestingly, in the case of SCORE, DAS, NoGAM, and DirectLiNGAM, we observe better stability of the FNR-$\hat{\pi}$, compared to the other methods, with the score matching-based approaches that are in general significantly better than the random baseline also with datasets of size 100.

**The effect of the graph size and density.** Figure 13 illustrates the F1 score and the FNR-$\hat{\pi}$ accuracy on datasets generated according to the vanilla scenario and ground truth graphs that differ in size and density. In particular, we consider the case of dense and sparse graphs, with $\{5, 10, 20, 50\}$ nodes. Interestingly, we see good stability of the F1 score across different graph dimensions in the sparse case. The decrease in performance due to larger graph sizes is more evident in the case of dense graphs: this is particularly true for dense graphs with 50 nodes, where the preliminary neighbours search step (described in Section C.1) before the CAM-pruning reduces the ability to

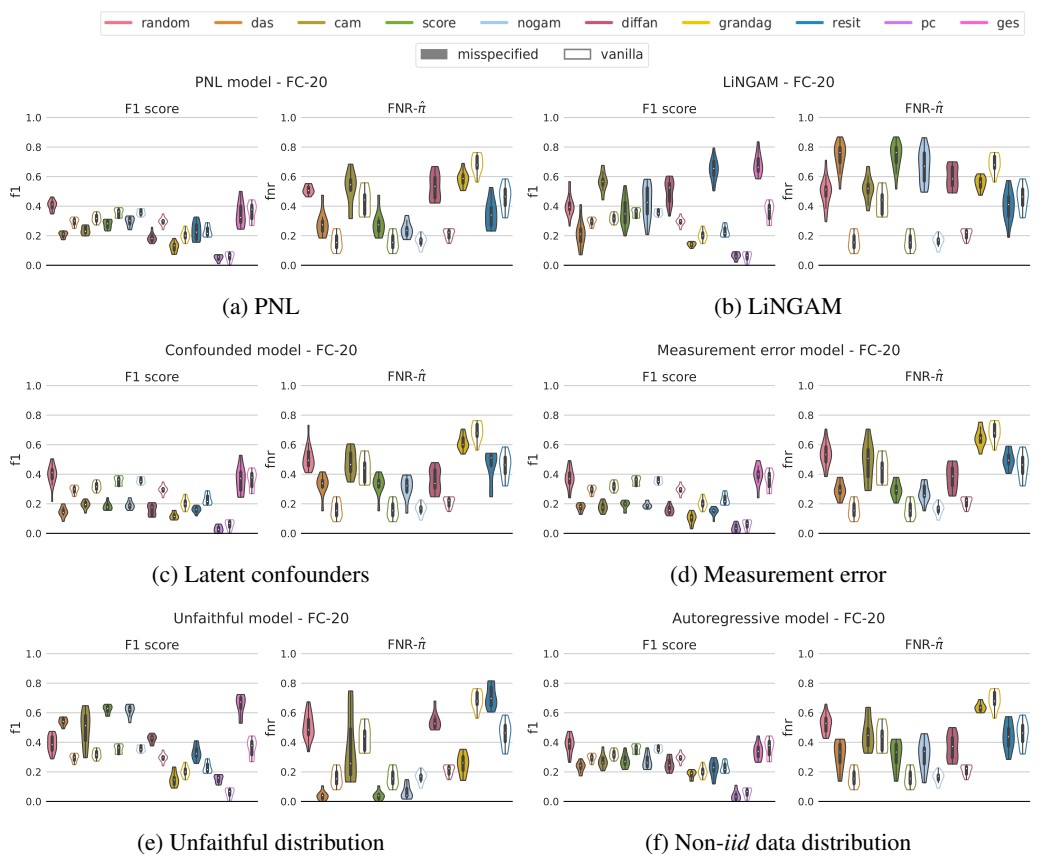

Figure 9: Experimental results on the misspecified scenarios. For each method, we also display the violin plot of its performance on the *vanilla* scenario with transparent color. F1 score (the higher the better) and FNR-$\hat{\pi}$ (the lower the better) are evaluated over 20 seeds on fully connected graphs with 20 nodes (FC-20). FNR-$\hat{\pi}$ is not computed for GES and PC methods, whose output is a CPDAG. Note that DirectLiNGAM performance does not appear, as both the linear mechanisms and non-Gaussian noise assumptions are violated.

infer true positives for most of the methods. Considering the FNR-$\hat{\pi}$ of the inferred orders, we see that, similarly to what we observed in the analysis of the F1 score, in the case of sparse ground truths most of the methods display stable results across different graph dimensions. Indeed, score matching-based approaches, as well as CAM, DiffAN, GraN-DAG, and DirectLiNGAM do not display any clear evidence of degraded performance for larger graphs. In the dense setting, instead, we see that CAM and DirectLiNGAM accuracy in the inference of the order is negatively affected by larger dimensionality.

**The balanced scoring function.** Figure 12 illustrates the inference accuracy of misspecified scenarios in terms of the Balanced Scoring Function (BSF), proposed by Constantinou [53]. This is defined as:

$$\frac{1}{2}\left(\frac{TP}{a} + \frac{TN}{i} - \frac{FP}{i} - \frac{FN}{a}\right),$$

where: *TP* and *FP* denote the true and false positives, respectively, and *TN* and *FN* are the true and false negatives; $a$ and $i$ represent the number of arcs and independencies in the true graph respectively. The key difference between the BSF and the F1 score is given by the fact that the balanced scoring function accounts for the whole confusion matrix (i.e. TP, FP, TN, FN), whereas in the F1 score the true negatives are not included in the computation. The BSF ranges from $-1$ (worst) to $1$ (best), with a value of $0$ corresponding to the score of an empty graph. In general, we expected the BSF to be correlated to the F1 score. The main difference that we see is that the F1 score in general penalizes PC and GES performance more than the BSF (when comparing their accuracy to the random baseline), meaning that they tend to infer graphs with low true positive rates and large true negative rates.

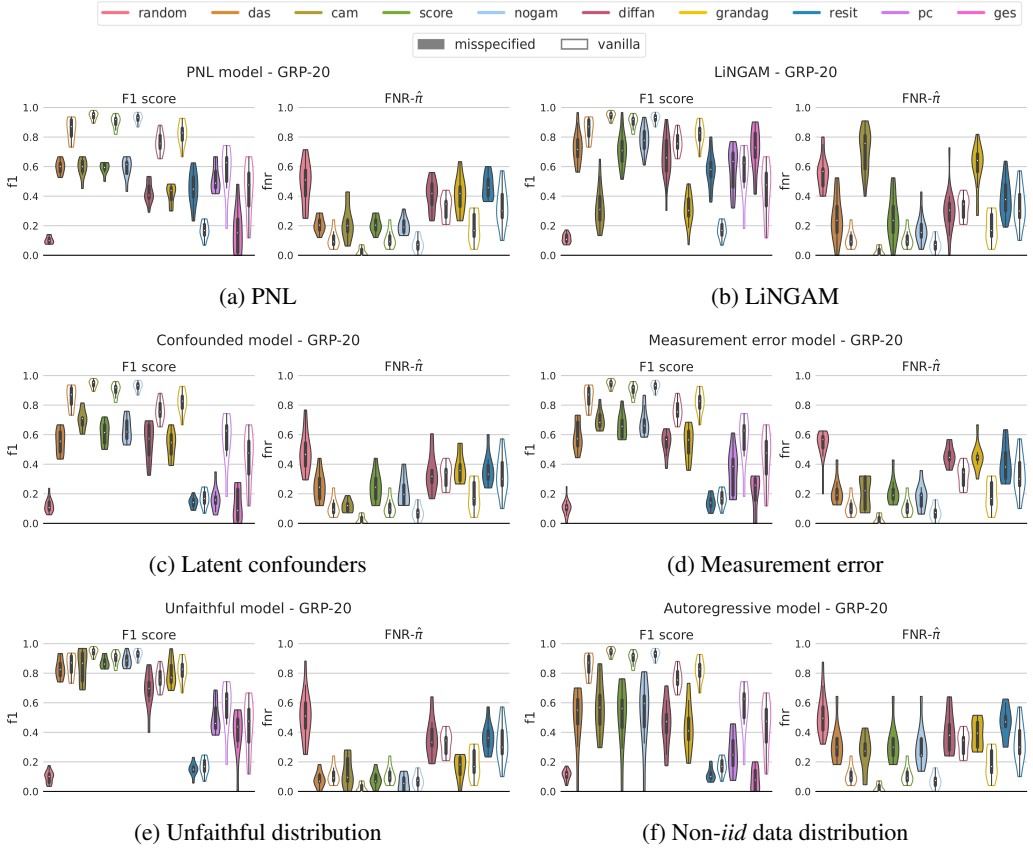

Figure 10: Experimental results on the misspecified scenarios. For each method, we also display the violin plot of its performance on the *vanilla* scenario with transparent color. F1 score (the higher the better) and FNR-$\hat{\pi}$ (the lower the better) are evaluated over 20 seeds on Gaussian Random Partitions graphs with 20 nodes (GRP-20). FNR-$\hat{\pi}$ is not computed for GES and PC methods, whose output is a CPDAG. Note that DirectLiNGAM performance does not appear, as both the linear mechanisms and non-Gaussian noise assumptions are violated.

## L    FCI experiments on confounded graphs

In this section, we describe the experimental setting for the FCI algorithm (Fast Causal Inference) [4]. Given that the method can handle latent confounders, we focus our experiments on data generated from graphs admitting latent common causes.

**PAG.** The output graphical object of FCI is a Partial Ancestral Graph (PAG) [47]. It admits six types of edges. We denote the two ends of an edge as *marks*. The possible marks are a tail ($-$), a circle ($\circ$), and an arrowhead ($>$), which combined allow for six edges. These graphs represent an equivalence class for *Maximal Ancestral Graphs*, which are graphical objects that represents the presence of confounders effects and selection bias [2].

**Metrics.** For the evaluation of the FCI inferred output, we adopt the strategy proposed by Heinze-Deml et al. [17] (see their Section 4.2). We define true positives, false positives, and false negatives over three possible adjacency matrices, each one defined by a specific query.

- *IsPotentialParent* query: the estimated adjacency matrix has $A_{ij} = 1$ if there is an edge between $X_i - X_j$, $X_i \multimap X_j$, $X_i \to X_j$, $X_i \circ\!\!- X_j$, $X_i \circ\!\!\multimap X_j$, $X_i \circ\!\!\to X_j$ in the estimated PAG, else $A_{ij} = 0$. $A_{ij} = 1$ denotes the case in which $X_i$ is a potential parent of $X_j$.

- *IsAncestor* query: the estimated adjacency matrix has $A_{ij} = 1$ if there is a path from $X_i$ to $X_j$ with edges of type $X_i \multimap X_j$, $X_i \to X_j$, $X_i \circ\!\!- X_j$ in the estimated PAG, else $A_{ij} = 0$. $A_{ij} = 1$ denotes the case in which $X_i$ is an ancestor of $X_j$.

- *IsPotentialAncestor* query: the estimated adjacency matrix has $A_{ij} = 1$ if there is a path from $X_i$ to $X_j$ with edges of type $X_i - X_j$, $X_i \multimap X_j$, $X_i \to X_j$, $X_i \circ\!\!- X_j$, $X_i \circ\!\!\multimap X_j$,

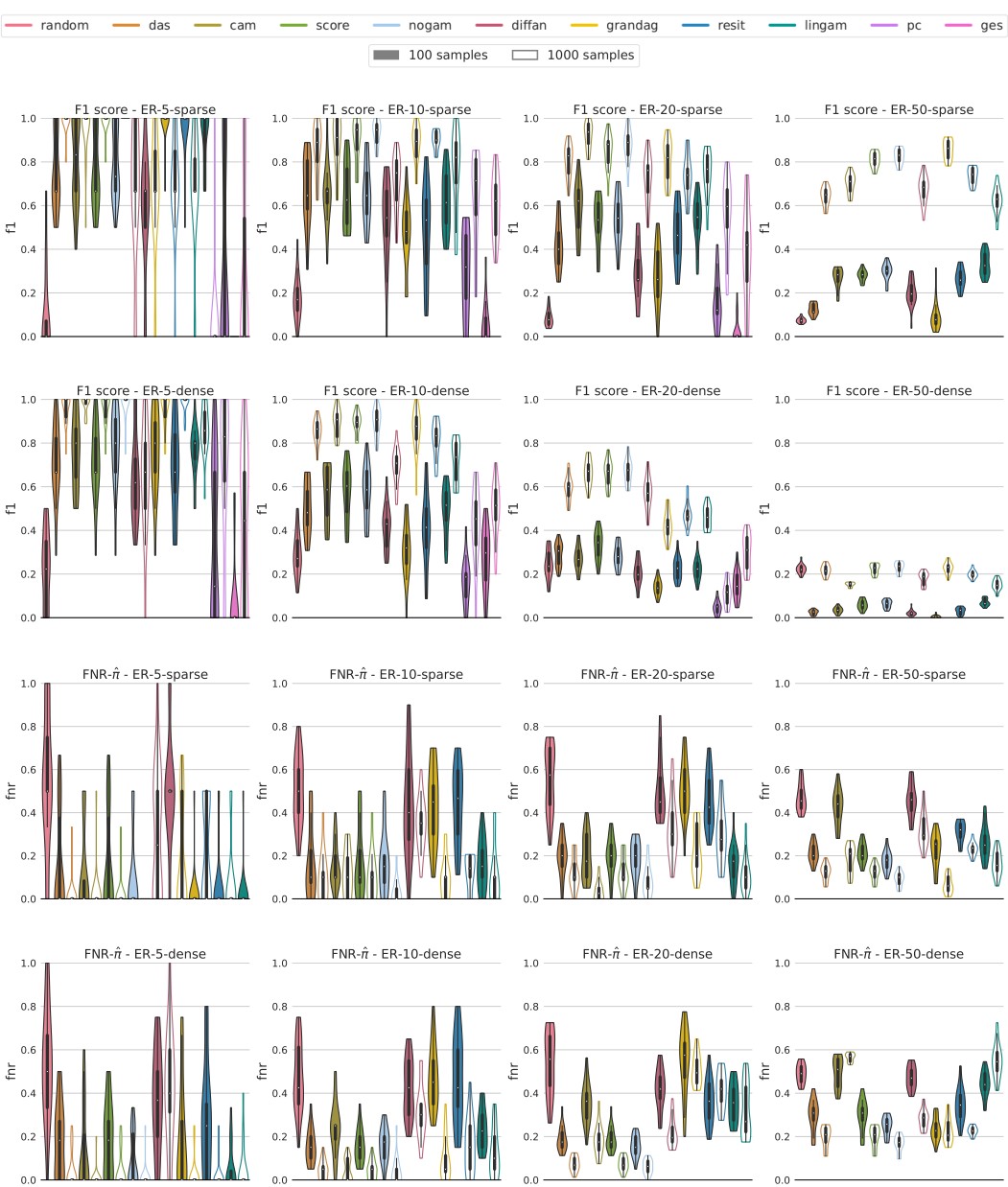

Figure 11: Experiments on the effect of the sample size. We compare the F1 score and FNR-$\hat{\pi}$ accuracy on datasets generated under the vanilla scenario with Gaussian noise, with different sample sizes. We remark that in the case of DirectLiNGAM, in order to provide meaningful results, we report the performance on datasets with non-Gaussian noise terms. Violin plots filled with color refer to datasets of size 100, and transparent violin plots refer to datasets of size 1000. The metrics are reported on Erdos-Renyi graphs of size $\{5, 10, 20, 50\}$ both in the sparse and dense case (PC and GES are not included for graphs of 50 nodes, as their computational demand is too high). Experiments are repeated over 20 different random seeds.

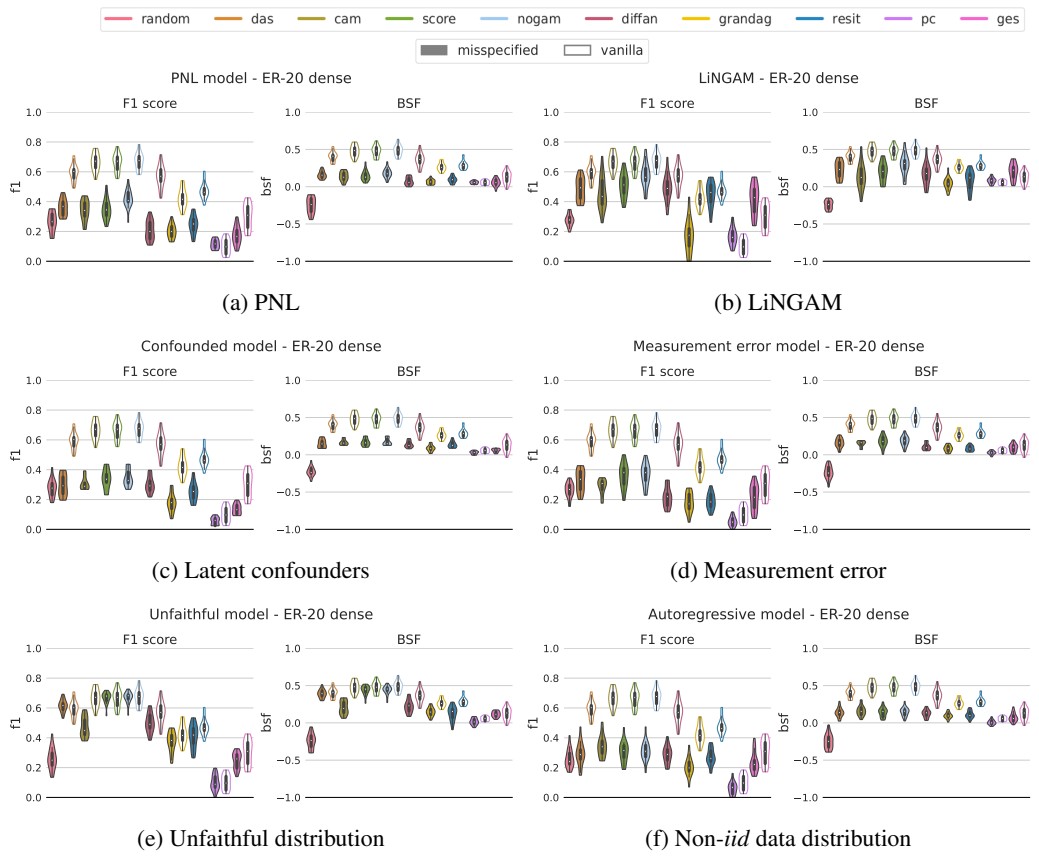

Figure 12: Experimental results on the misspecified scenarios. For each method, we also display the violin plot of its performance on the *vanilla* scenario with transparent color. BSF and F1 score (the higher the better) are evaluated over 20 seeds on Erdos-Renyi dense graphs with 20 nodes (ER-20 dense).

$X_i \circ\!\!\rightarrow X_j$ in the estimated PAG, else $A_{ij} = 0$. $A_{ij} = 1$ denotes the case in which $X_i$ is a potential ancestor of $X_j$.

For each adjacency matrix defined by one of the three queries, we define true positives, false negatives, and false positives as follows:

- A true positive (TP) is a pair $i, j$ with $A_{ij} = 1$ in both the inferred and ground truth adjacency matrices (with the ground truth DAG converted to a PAG).

- A false negative (FN) is a pair $i, j$ with $A_{ij} = 0$ in the inferred matrix, and $A_{ij} = 1$ in the ground truth (with the ground truth DAG converted to a PAG).

- A false positive (FP) is a pair $i, j$ with $A_{ij} = 1$ in the inferred matrix, and $A_{ij} = 0$ in the ground truth (with the ground truth DAG converted to a PAG).

Given these definitions of TP, FN, FP, we define the F1 score as $F1 = \frac{TP}{TP+0.5(FP+FN)}$, which we use to present our empirical results in Figure 14.

## M Deep-dive in PC and GES experimental results

In this section, we analyze GES and PC performance in terms of false positive and false negative rates on graphs characterized by different numbers of nodes and edge density. In Section 4.1.1, we discuss the case of inference with PC and GES on Erdös-Renyi dense and *large* graphs (20 nodes): Figure 1 in the main text reports PC and GES F1 score to be consistently and significantly worse than random across all of the tested scenarios. We argue that this is in line with previous findings in the literature

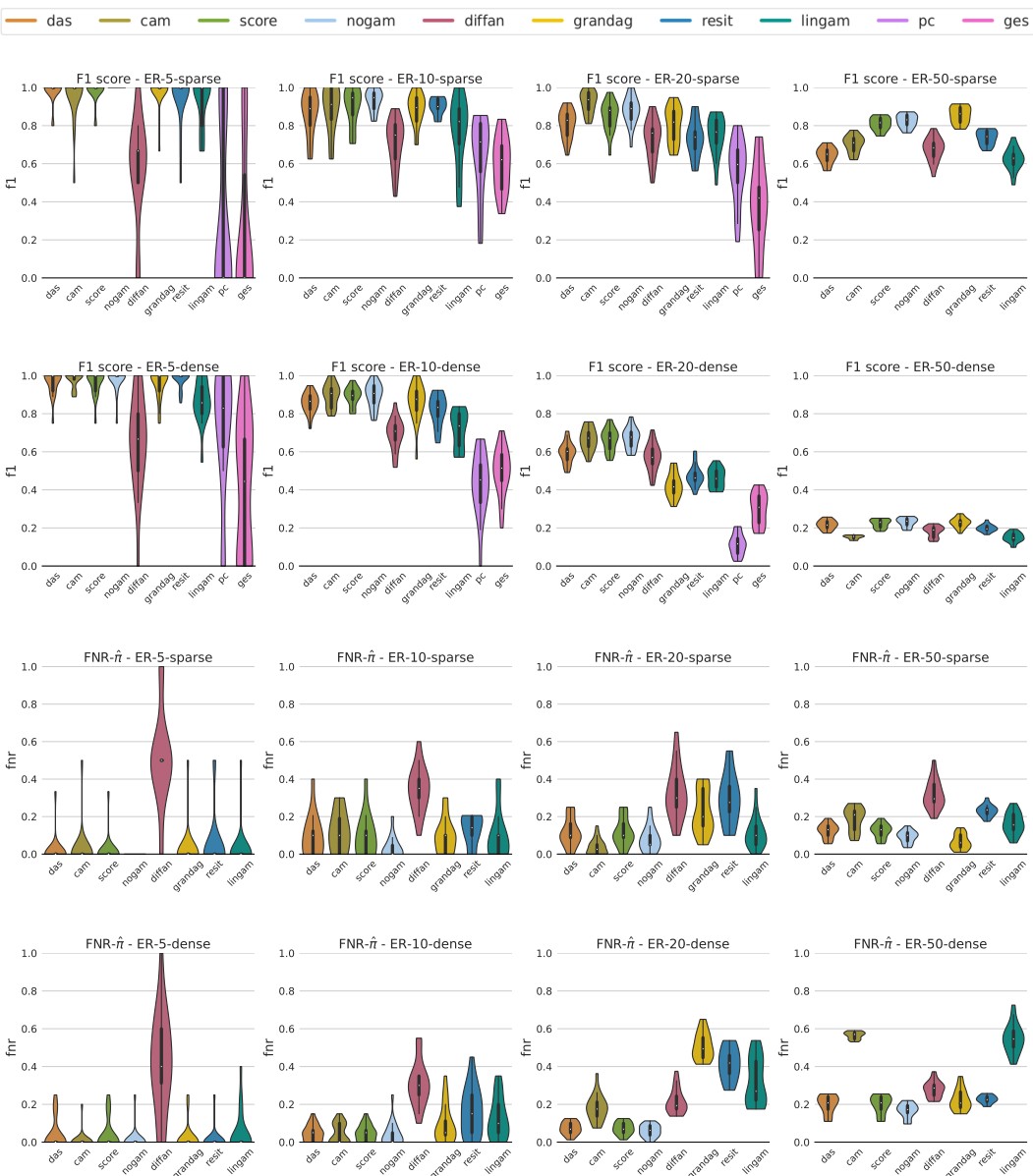

Figure 13: Experiments on the effect of the graph size and graph density. We compare the F1 score and FNR-$\hat{\pi}$ accuracy on datasets generated under the vanilla scenario with Gaussian noise, on ground truth graphs with the number of nodes $\{5, 10, 20, 50\}$ both in the sparse and dense case (PC and GES are not included for graphs of 50 nodes, as their computational demand is too high). We remark that in the case of DirectLiNGAM, in order to provide meaningful results, we report the performance on datasets with non-Gaussian noise terms. The metrics are reported on Erdos-Renyi graphs. Experiments are repeated over 20 different random seeds.

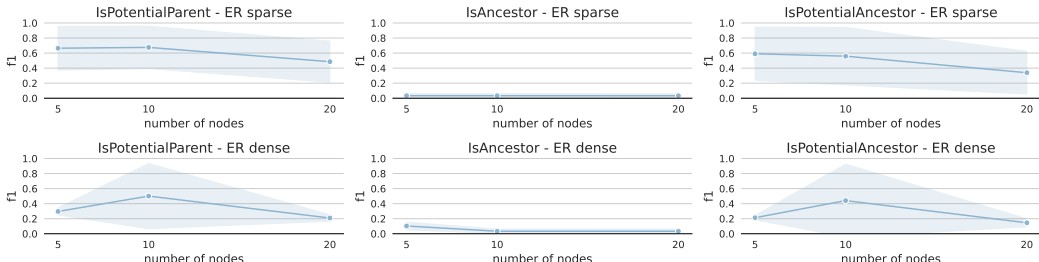

Figure 14: FCI performance on dense and sparse ER graphs, on datasets generated under latent confounders effects.

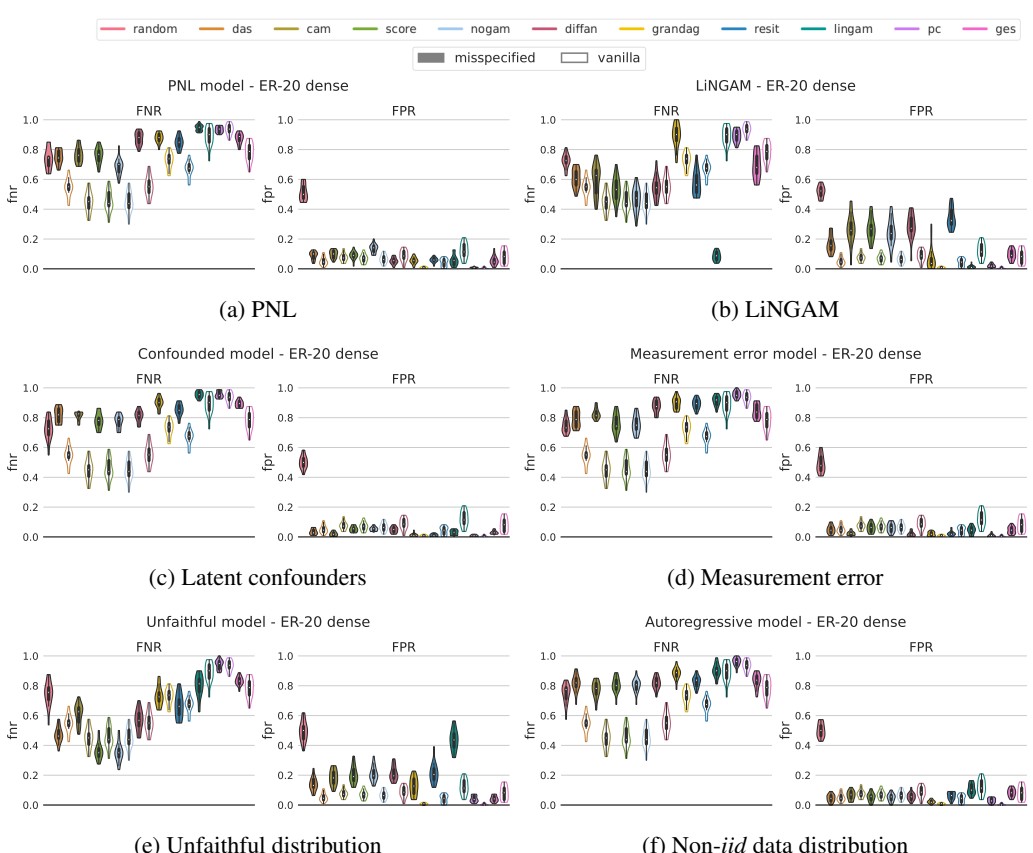

Figure 15: FNR (False Negative Rate) and FPR (False Positive Rate) of the experiments on the misspecified scenarios, on Erdos-Renyi dense graphs with 20 nodes (ER-20 dense). For each method, we also display the violin plot of its performance on the vanilla scenario with transparent color. The noise terms are normally distributed, except for the LiNGAM model, in which case we generate disturbances according to a non-Gaussian distribution.

by Uhler et al. [7], which shows that the set of distributions that are not strong-faithful ([54]) has non-zero Lebeasgue measure, in contrast to the set of unfaithful distributions that has zero measure (which justifies the common assumption of faithful causal models). In particular, the measure of the set of not strong-faithful distributions tends to increase for large and dense causal graphs, which we argue is key to explain the degraded performance of PC in the *ER-20 dense* graphs of Figure 1. Coherently with our claim, Figures 19 and 20 show FNR and FPR consistently better than random for both PC and GES, respectively on sparse and dense graphs of 5 nodes.

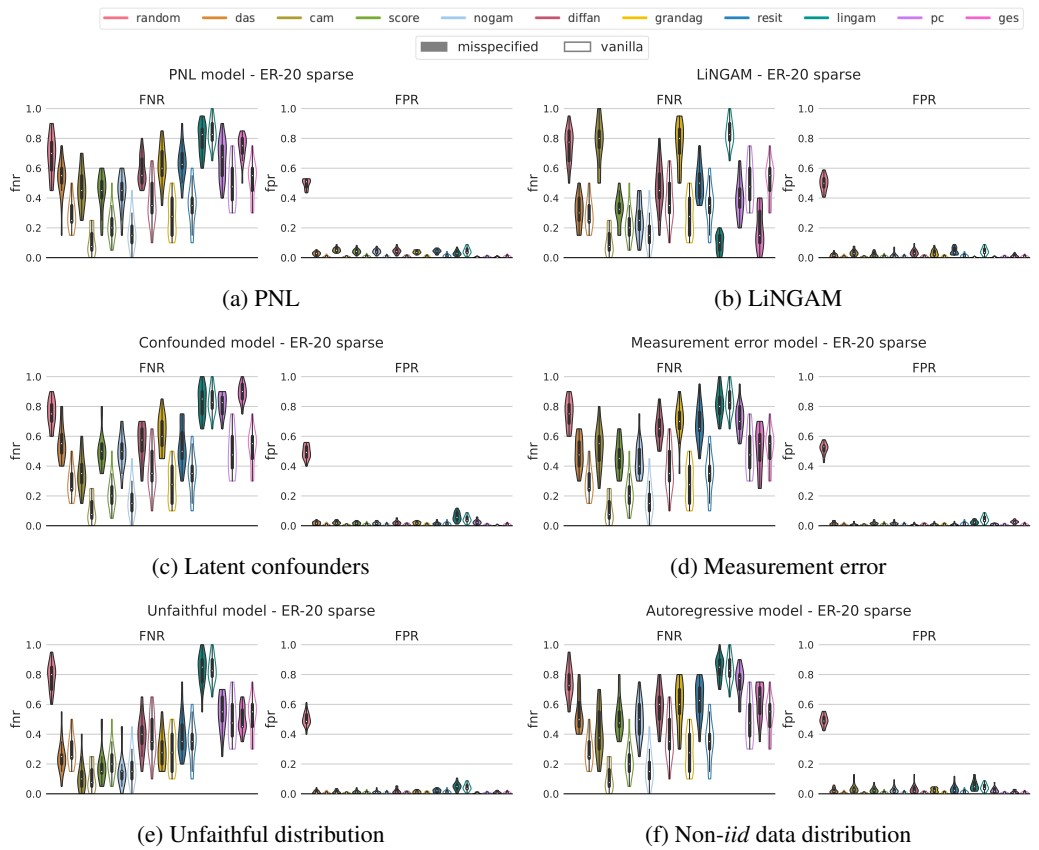

Figure 16: FNR (False Negative Rate) and FPR (False Positive Rate) of the experiments on the misspecified scenarios, on Erdos-Renyi sparse graphs with 20 nodes (ER-20 sparse). For each method, we also display the violin plot of its performance on the vanilla scenario with transparent color. The noise terms are normally distributed, except for the LiNGAM model, in which case we generate disturbances according to a non-Gaussian distribution.

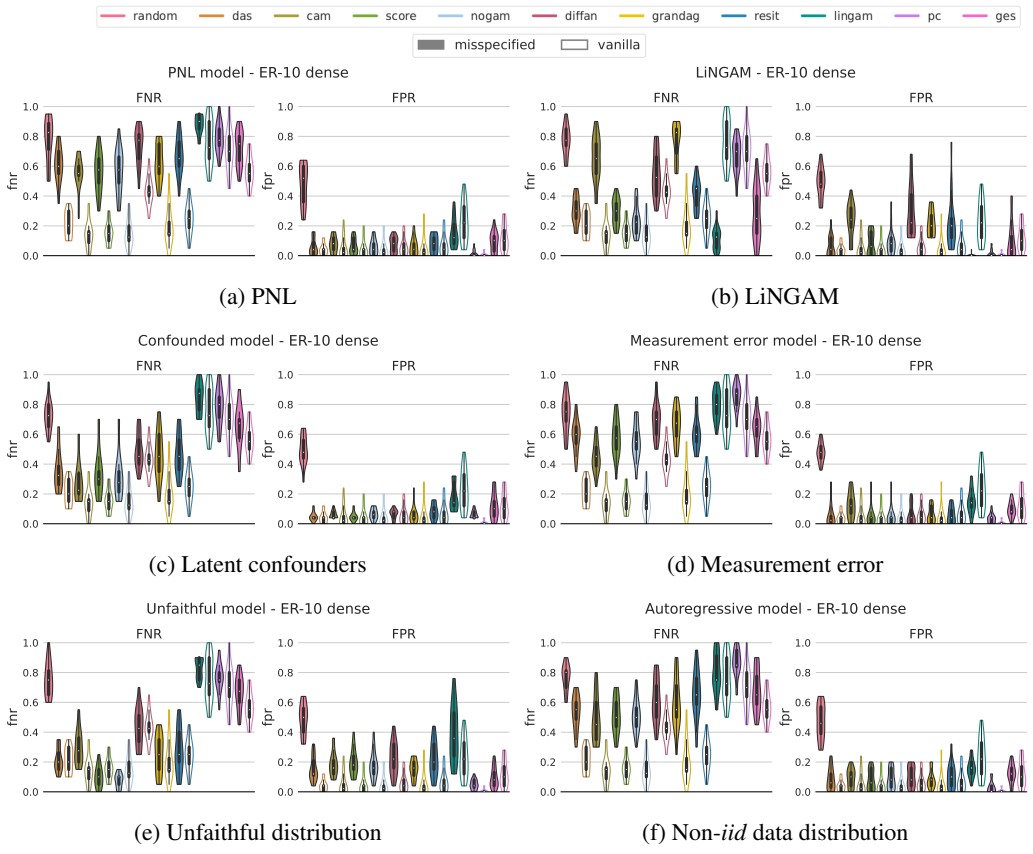

Figure 17: FNR (False Negative Rate) and FPR (False Positive Rate) of the experiments on the misspecified scenarios, on Erdos-Renyi dense graphs with 10 nodes (ER-10 dense). For each method, we also display the violin plot of its performance on the vanilla scenario with transparent color. The noise terms are normally distributed, except for the LiNGAM model, in which case we generate disturbances according to a non-Gaussian distribution.

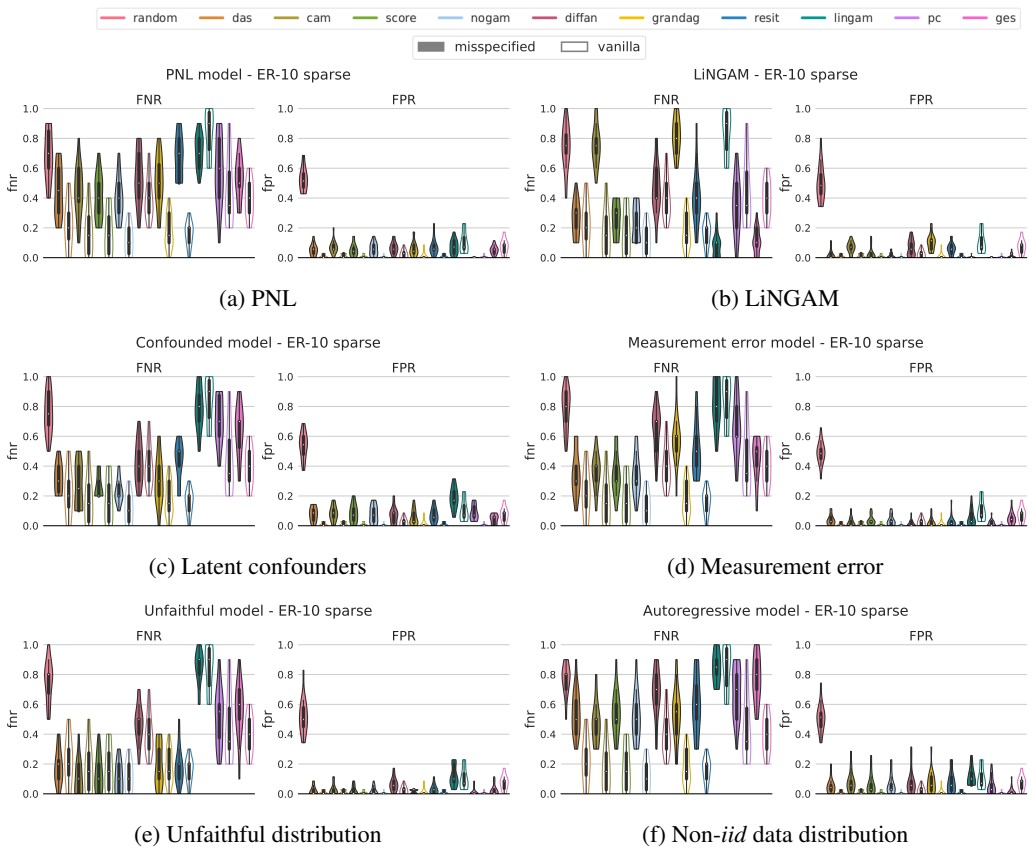

Figure 18: FNR (False Negative Rate) and FPR (False Positive Rate) of the experiments on the misspecified scenarios, on Erdos-Renyi sparse graphs with 10 nodes (ER-10 sparse). For each method, we also display the violin plot of its performance on the vanilla scenario with transparent color. The noise terms are normally distributed, except for the LiNGAM model, in which case we generate disturbances according to a non-Gaussian distribution.

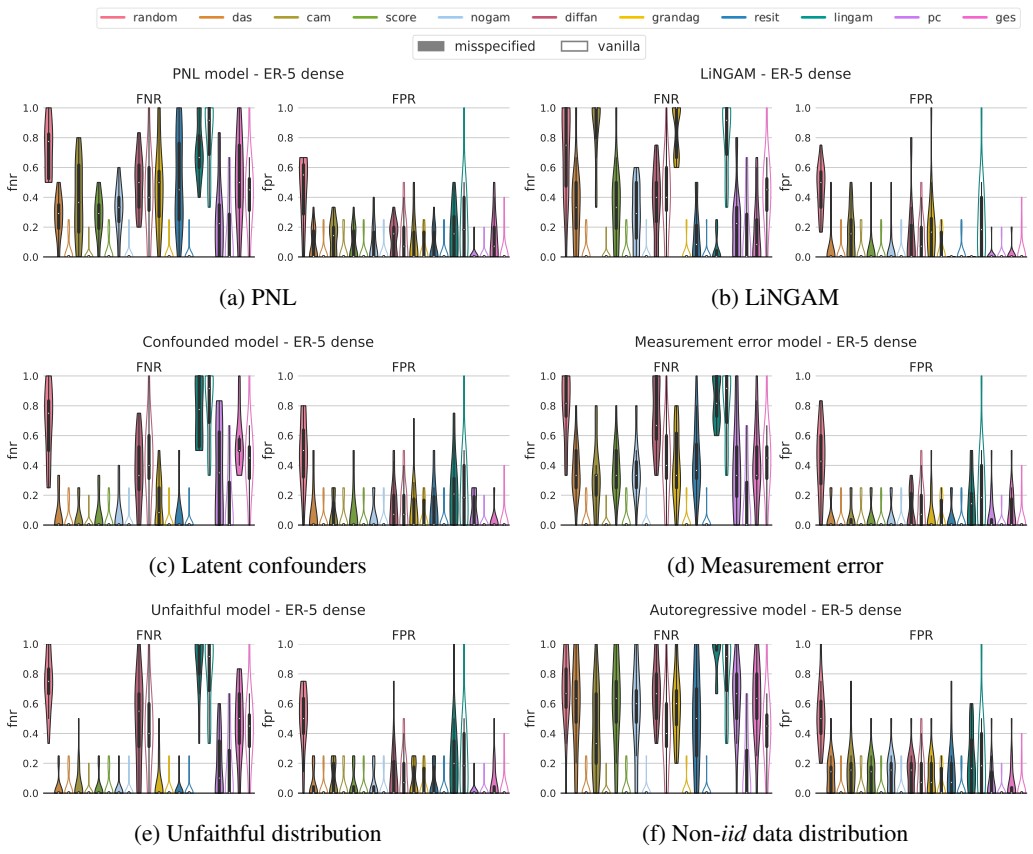

Figure 19: FNR (False Negative Rate) and FPR (False Positive Rate) of the experiments on the misspecified scenarios, on Erdos-Renyi dense graphs with 5 nodes (ER-5 dense). For each method, we also display the violin plot of its performance on the vanilla scenario with transparent color. The noise terms are normally distributed, except for the LiNGAM model, in which case we generate disturbances according to a non-Gaussian distribution.

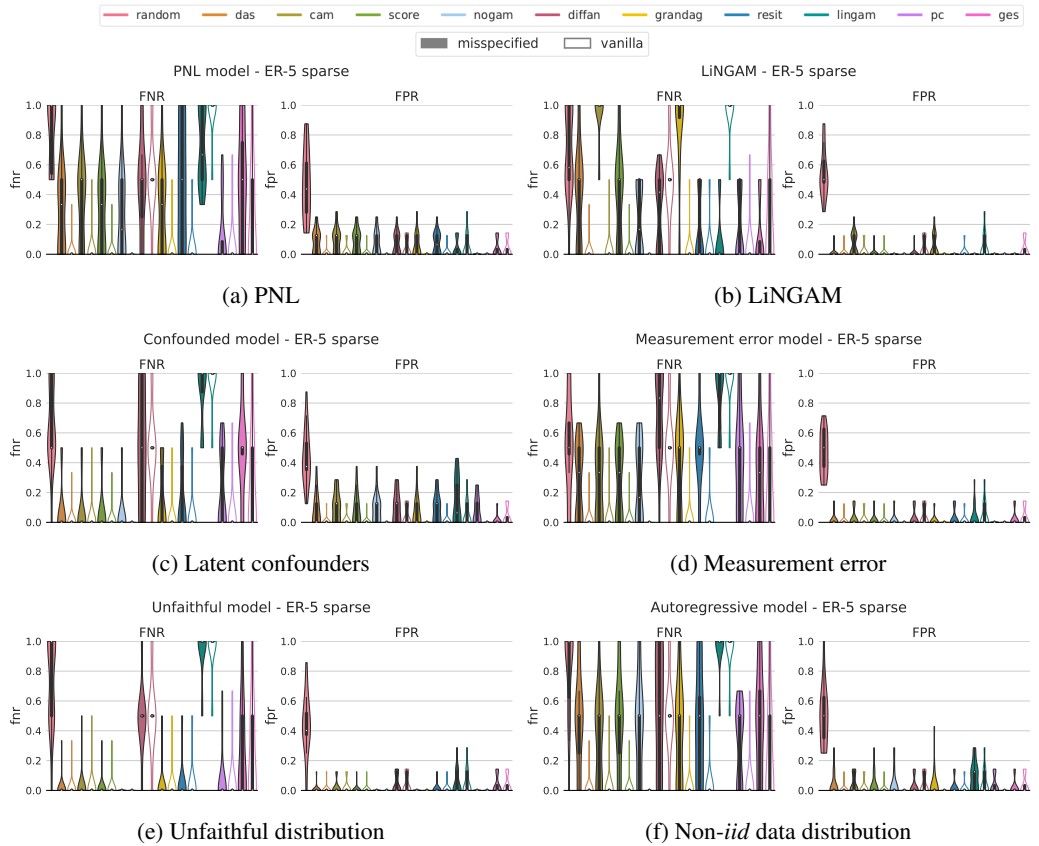

Figure 20: FNR (False Negative Rate) and FPR (False Positive Rate) of the experiments on the misspecified scenarios, on Erdos-Renyi sparse graphs with 5 nodes (ER-5 sparse). For each method, we also display the violin plot of its performance on the vanilla scenario with transparent color. The noise terms are normally distributed, except for the LiNGAM model, in which case we generate disturbances according to a non-Gaussian distribution.

