# OpenReview forum: "Assumption violations in causal discovery and the robustness of score matching"
_NeurIPS.cc/2023/Conference — NeurIPS 2023 poster_

### Official Review · Reviewer_JGo2 · 2023-07-06

**Soundness:** 3 good
**Presentation:** 4 excellent
**Contribution:** 3 good
**Rating:** 8
**Confidence:** 4

**Summary:**

The authors focus on evaluating causal structure learning algorithms.  These algorithms, in order to claim that their outputs are causal, require various assumptions, most of which can't be tested easily in observational data.  This work aims to test how robust a set of causal structure learning algorithms are to violations of these assumptions.  The authors pay particular attention to a group of additive noise model-based methods that use score matching and find that they are generally the most robust to assumption violations.

**Strengths:**

I appreciate the authors' focus on evaluating assumptions - this is rarely done and is very important for the field.

The assumption-violation settings are well-chosen, and Table 1 provides a helpful reference.  In general, the way you chose to implement these seems reasonable.

The random baseline, as well as showing both the misspecified and vanilla results side-by-side, is a great choice and helps readability of otherwise dense graphics.

**Weaknesses:**

The diversity of method is lacking.  In Table 1, FCI and DirectLiNGAM are included, but neither are present in the actual evaluation.  This means that of the 9 methods presented in the evaluation, 7 are additive noise model-based methods with nearly the same assumptions.  While it says that experiments on FCI are in Appendix I.5, no section I.5 exists.  Appendix L looks like it starts discussing FCI results but never actually does so.  Similarly, the Figure 1 caption says that DirectLiNGAM results are in Appendix I.4, but again, no section I.4 exists.  I understand that it's much easier to compare algorithms that all make the same assumptions, since they can be compared against the same type of baseline.  But the result is that it's more a comparison of which additive noise-based method is more robust to assumption violations than a true comparison across structure learning methods.  I otherwise think this work is great, but for a paper focused solely on evaluation and pitched as a comparison across different types of methods (since the conclusion is that score matching methods are best), it really feels like it's lacking substantial representation from those other types of methods.

I understand you're already manipulating a lot of settings in combination, but I wish the baseline datasets were more functionally varied.  From what it looks like, since all of the methods (apart from LiNGAM, which was omitted from the evaluation) require nonlinearity, you defined that nonlinear mechanism as a Gaussian process.  However, there are other nonlinear mechanisms that exist, and some additional variations there would help round out the analysis.


**Questions:**

You discuss SCORE, NoGAM, and DiffAN as being part of the same class of methods that derive constraints on the score function (in Section 3.2.1).  However, in the post-results discussion (Section 4.1.1), you discuss the group of SCORE, DAS, and NoGAM instead.  Despite operating similarly to NoGAM and SCORE, why do you think DiffAN was not as robust?  And similarly, what makes DAS comparable here, despite not being part of the original score-based discussion?

Following on to my comment about using a Gaussian process as the nonlinear mechanism, what led to this choice?  Did you consider and reject other mechanisms, or do you have reason to believe that this one is realistic enough to be a reasonable choice?

**Limitations:**

In general, this work is focused on addressing limitations in the field (since most work merely specifies the assumptions without assessing empirically how necessary these assumptions are).  I think the results of this work are a bit limited, as discussed above, due to the limited diversity of methods assessed.

In terms of social impact, I only see positives to deeper exploration of the robustness (or brittleness) of methods in the face of assumption violations.

---

> ### Author Rebuttal · Authors · 2023-08-09
>
>
> We thank the reviewer for considering our work and for their valuable comments and insights.
>
> -   > The diversity of method is lacking. In Table 1, FCI and DirectLiNGAM are included, but neither are present in the actual evaluation.
>
>
> 	In the case of FCI, when confounders are not present in the model, FCI is reduced to simple PC. The only interesting information comes in the case of data generated under confounders effects: in this setting, FNR, FPR, and F1 definitions of section 3.2 (paragraph starting at L204) don’t allow for straightforward comparison of PAGs with DAGs and CPDAGs. This is why we rely on the metrics used in Heinze et al., 2017. In this sense, a comparison with the other methods’ robustness is not discussed, as we would be comparing different metrics, which is not immediate.
>
> 	In the case of DirectLiNGAM, we omit the results from the main manuscript as most of the data are generated from nonlinear models with Gaussian noise, which is heavily incompatible with the method's assumptions (in practice we observed DirectLiNGAM performance to be significantly worse than the random baseline in this setting, which is not surprising). For this reason, we moved the discussion on DirectLiNGAM to Appendix I.2 _Experiments under arbitrary distribution of the noise terms_.
>
> 	Concerning the reviewer's remark that _diversity of methods is lacking_, we agree that our analysis is biased towards methods based on nonlinear additive noise models, and we agree to be more explicit about this fact in the camera ready version of the paper. Nevertheless, our benchmark also includes the most prominent constraint and score-based methods in literature that also have an available and well maintained Python implementation. In this regard, please consider the well established library for causal discovery _causal-learn_: the only algorithms available for constraint and score based methods are PC, FCI, GES, and additionally DirectLiNGAM. Similarly, for the _Dodiscover_ library that we used in our work. Limitations in the available Python implementations are substantial and thus reflected in our benchmark, where existing implementations were privileged. In this sense, we believe that our software contribution is an additional important value of our work.
>
> -   > While it says that experiments on FCI are in Appendix I.5, no section I.5 exists … Similarly, the Figure 1 caption says that DirectLiNGAM results are in Appendix I.4, but again, no section I.4 exists.
>
>
> 	We now notice that we introduced a mismatch in the section’s numbers between the file with only the main manuscript with respect to the file with the supplementary material. We are sorry for the confusion this has generated and ask please to consider appendix I.2 of the pdf with supplementary material for the discussion on DirectLiNGAM performance. The analysis of FCI is in Appendix L (as already found out by the reviewer) rather than Appendix I.5. Again, we apologize for the inconvenience.
>
> -   > you defined that nonlinear mechanism as a Gaussian process. However, there are other nonlinear mechanisms that exist, and some additional variations there would help round out the analysis.
>
>
> 	and again, in the _Questions_ section:
>
> 	> Following on to my comment about using a Gaussian process as the nonlinear mechanism, what led to this choice? ...
>
> 	As correctly hinted by the reviewer, this choice is mostly due to the combinatorial nature of the number of experiments (“I understand you are already manipulating a lot of settings"), which _explodes_ each time a new _dimension_ for data generation is added. In addition to this, we point to a large amount of established and recent literature using our same proposal in the data generation (e.g. _Distinguishing cause from effect using observational data: methods and benchmarks_, Mooij et al. 2009; _Causal discovery with reinforcement learning_, Zhu et al., 2016, + the papers introducing SCORE, DAS, NoGAM, GraN-DAG, RESIT, CAM, ...).
>
> -   > … Despite operating similarly to NoGAM and SCORE, why do you think DiffAN was not as robust?
>
>
> 	We discuss this point in L 246-248 “… the difference is in that it [DiffAN] estimates the score function with probabilistic diffusion models, whereas SCORE, NoGAM, and DAS rely on score matching estimation”. The main novelty of DiffAN with respect to SCORE, is that it estimates the score function with probabilistic diffusion models, which appears to be less effective than score-matching in the finite samples regime. To make this point, we dedicate the content of Section 4.2, with title _Is the choice of statistical estimators neutral?_. In particular, see L 355-358.
>
> -   > ... what makes DAS comparable here, despite not being part of the original score-based discussion?
>
>
> 	The details of DAS are discussed in Appendix C.9, in which we clarify that DAS can be seen as an accelerated version of SCORE, where they share the same ordering procedure (based on identifying leaf nodes on the variance of the Jacobian of the score function estimated via score matching), while they differ in the pruning step (DAS also has a preliminary pruning step that exploits information in the Jacobian of the score function to reduce the computational cost of running CAM pruning afterward). We will include this discussion in the main paper in Section 3.2.1.

---

> > ### Comment · Reviewer_JGo2 · 2023-08-15
> >
> > I thank the authors for their thoughtful reply.  I still think this is a valuable contribution, despite the limitations.  Having published before in this space (evaluating causal modeling algorithms), I'm well aware of the explosion of the experimental combinations you get when you're trying to be thorough, and the authors' justifications for omitted experiments are reasonable.  This sort of paper is generally very hard to get published just because of all of the possible alternate experiments that can be run, so I'm certainly not going to hold it against the authors for failing to run every conceivable combination.  With the proposed clarifications, I'm happy to raise my score.

---

### Official Review · Reviewer_eJF4 · 2023-07-06

**Soundness:** 3 good
**Presentation:** 3 good
**Contribution:** 3 good
**Rating:** 6
**Confidence:** 4

**Summary:**

The authors present the results of an extensive empirical study evaluating the performance of algorithms for learning the structure of causal graphical models from observational data. In particular, the study examines the robustness of these algorithms to violations of various untestable assumptions that are typically made by one or more of these algorithms. These violations are Gaussian noise, non-Gaussian noise, linear mechanisms, nonlinear mechanisms, unfaithful distributions, confounding effects, and measurement errors.

**Strengths:**

The focus of the paper is extremely important and overdue. There is only very weak understanding with in the research community about how existing methods for learning causal structure respond in practice to various types of misspecification. This is particularly true for several methods that have been devised fairly recently.

The experiments are extensive. As the authors state: “Our experimental protocol consists of more than 2M experiments with 11 different causal discovery methods on more than 60000 datasets synthetically generated.”

The authors release code for both the data generation and for the methods themselves.

The authors provide a random baseline, which is very useful for contextualizing the results. In some cases (e.g., confounding, autoregressive), the condition reduces almost all methods to essentially random performance.

The results are examined in some detail, and the authors provide a variety of high-level takeaways to help readers interpret the vast number of experiments that they have performed.

**Weaknesses:**

The authors use false-positive and false-negative rates (and F1) as their measures of error. This ignores a great deal of recent scholarship [1, 2, 3, 4] regarding the problems with these measures and proposals of better measures. Some of the measures require model parameters [3] or knowledge of likely intervention targets and outcome measures [4], but others do not [1, 2]. The paper would be improved by examining the results of these error measures as well as the simpler ones of FPR, FNR, and F1.

The authors observations regarding the results for unfaithful models provide a specific case of one reason for using these alternative measures. In an effort to explain why faithfulness violations had little effect on most methods, they say: “…the set of edges of the graph G ̃ is sparser than that of the ground truth, due to the cancellation of causal effects. Thus, given that inference on sparser graphs is generally easier, it can positively affect the empirical performance, in line with our observations.” As Constantinou [2] points out, graph sparsity directly affects the F1 score, and his proposed BSF accounts for this.

The simplest way to summarize the results (at least those in Figure 1) is that the relative ordering of performance of the methods is maintained under misspecification. That could reflect greater robustness of the score-matching based methods. However, it could equally well reflect just overall performance or some bias in the particular error measures used to evaluate the learned structures (that’s one reason for the comment above about error measures). Despite this, the authors claim that the results demonstrate that specific methods are surprisingly robust (“Our empirical findings indicate that score matching-based methods are surprisingly capable of partial recovery of the graph structure in several of the misspecified scenarios.”).

In the “Implications” section (312-315), the authors correctly point out that “Our experimental findings show that most of the benchmarked methods significantly decrease their performance on the misspecified models. This is particularly problematic since the violations considered in this work are realistic and met on many real-world data.” This is correct and, to my mind, this is one of the primary results of the paper. Almost without exception, each of these misspecifications reduces the performance of nearly all methods. However, the authors then go on to note: “On the other hand, we observe surprising robustness in the inference of score matching-based methods.” This is much less well supported by the empirical results. The paper would be greatly improved by providing empirical results that distinguished between the hypotheses of “consistent proportional decrease from varying initial performance” and “special robustness of some methods despite their initial high performance”. I’m seeing strong evidence for the first, but not for the second.

There are at least two aspects of the results that are not sufficiently explained and should be. First, the paper should explain why PC generally performs *worse than random* in terms of F1. Random or near-random performance is one thing, but performance that is consistently worse than random needs to be explained. Related to this, why do misspecifications such as measurement error further *decrease* the F1 score of PC?  We would expect that measurement error would push the F1 score more toward random rather that further decrease performance.

Second, the authors should explain why misspecification sometimes *improves* other scores (e.g., RESIT FNR-pi under Autoregressive, GES F1 score under LiNGAM). In one case (RESIT FNR-pi under PNL) you do explain (273-275), but the other anomalies are not explained.

The results of the hypothesis tests against the random baseline should be displayed on Figure 1 somehow.

*References*

1. Constantinou, A. C., Liu, Y., Chobtham, K., Guo, Z., and Kitson, N. K. (2021). Large-scale empirical validation of Bayesian Network structure learning algorithms with noisy data. *International Journal of Approximate Reasoning*, vol. 131, pp. 151-188.

2. Constantinou, A. C. (2019). Evaluating structure learning algorithms with a balanced scoring function. *arXiv preprint arXiv:1905.12666*.

3. Gentzel, A., Garant, D., & Jensen, D. (2019). The case for evaluating causal models using interventional measures and empirical data. *Advances in Neural Information Processing Systems*, *32*.

4. Peters, J., & Bühlmann, P. (2015). Structural intervention distance for evaluating causal graphs. *Neural computation*, *27*(3), 771-799.

**Questions:**

(Repeating from above)

1. Why does PC generally perform *worse than random* in terms of F1. Random or near-random performance is one thing, but performance that is consistently worse than random needs to be explained.

2. Why do misspecifications such as measurement error further *decrease* the F1 score of PC?  We would expect that measurement error would push the F1 score more toward random rather that further decrease performance.

3. Why does misspecification sometimes *improves* other scores (e.g., RESIT FNR-pi under Autoregressive, GES F1 score under LiNGAM). In one case (RESIT FNR-pi under PNL) you do explain (273-275), but the other anomalies are not explained.

**Limitations:**

(none)

---

> ### Author Rebuttal · Authors · 2023-08-09
>
> We thank the reviewer for their effort spent in providing valuable feedback on our work. In the response, we will follow the paragraph listing structure of the review.
>
> -   > The authors use false-positive and false-negative rates (and F1) as their measures of error. This ignores a great deal of recent scholarship [1, 2, 3, 4] ...
>
> 	We agree that [1, 2] Balanced Score Function (BSF) proposal seems appropriate for the evaluation of causal graphs and it is appealing in the sense that requires no extra knowledge about model parameters or interventions (which are out of scope in our work). Still, we argue that F1 is a valuable choice for the evaluation of the causal graph outputs. In the rebuttal's `.pdf` file (Figure 2), we propose example plots that show a strong correlation between F1 and BSF. This is in line with the results found in Figure 1 of [2]: in the comparison between F1 and the normalized BSF, we see perfect consistency between the two scores in all of the proposed scenarios, except for one (scenario 3.3 on the x-axis). This suggests that using BSF rather than F1 score would not revert any of the conclusions about the _robustness_ of the involved methods with respect to model misspecification, which is the core of our work. We support our claim with empirical evidence by comparing the BSF and the F1 score (Figure 3 in the rebuttal `.pdf`) in the case of unfaithful data (mentioned in the second paragraph of the _Weaknesses_ section by the reviewer): BSF and F1 score follows a consistent pattern in the performance, meaning that our conclusions in this setting are not biased by the choice of F1 as one of the metrics. We notice that the normalized BSF is always larger than F1, which can be explained by the inclusion of the TN in the count of the score, and that is consistent with the observations in Figure 1 of [2].
>
> 	As the choice of the scoring function in causal discovery is far from trivial, we agree that BSF results would be a valuable addition to our work, as they provide a summary metric accounting for the entire confusion matrix. We commit to add a discussion on BSF results in the appendix.
>
> -   > The simplest way to summarize the results is that the relative ordering of performance of the methods is maintained under misspecification. That could reflect greater robustness of the score-matching based methods. However, it could equally well reflect just overall performance or some bias in the particular error measures used ...
>
> 	Please refer to the comment above (first point in the response)
>
> -   > ... However, the authors then go on to note: “On the other hand, we observe surprising robustness in the inference of score matching-based methods.” This is much less well supported by the empirical results. ...
>
> 	Please refer to the second point in the common response.
>
> -   > The paper would be greatly improved by providing empirical results that distinguished between the hypotheses of “consistent proportional decrease from varying initial performance” and “special robustness of some methods despite their initial high performance”.
>
> 	Please refer to the second point in the common response, especially the remark we make about the use of the word robustness: as it is common in machine learning, we use it as a synonym for performance in misspecified scenarios. If the use of the word robustness sounds confusing here, we are willing to replace it with the expression _performance under misspecification_.
>
> -   > … the paper should explain why PC generally performs _worse than random_ in terms of F1.
>
> 	Please refer to the first point in the common response. We commit to add the discussion proposed also in the main text, as this is indeed of interest.
>
> -   > …Related to this, why do misspecifications such as measurement error further _decrease_ the F1 score of PC? …
>
> 	We point the reviewer to the closed form example in Section 2 of _Causal discovery in the presence of measurement error: Identifiability conditions,_ Zhang et al., 2017 : given the causal graph X_1 ← X_2 → X_3 with variables subject to measurement error and to linear mechanisms, it is possible to show that the correlation between X_1 and X_2 vanishes as the noise-signal ratio tends to infinity (implying an increase in the false negatives rate when using correlation to test for conditional independence). Inversely, checking the correlation coefficient between X_1 and X_3 conditional on X_2, as the noise-signal ratio goes to infinity this tends to approximate the correlation coefficient between X_1 and X_2 in the absence of measurement error (i.e. we would observe an increase in the false positives rate).
>
> 	This intuition is corroborated by the observations in Figures 14d to 18d in Appendix, which report the FNR and FPR.
>
> -   > authors should explain why misspecification sometimes _improves_ other scores (e.g., RESIT FNR-pi under Autoregressive, GES F1 score under LiNGAM).
>
> 	In the case of GES, the LiNGAM model satisfies the assumption of the method, thus we have no reason to expect the performance to be negatively affected.
>
> 	In the case of RESIT in the autoregressive setting, the discrepancy in performance on vanilla and non-iid data is substantially equal to that observed in all of the other misspecified scenarios (Figure 1 b,c,d,e right), while the only remarkable gap is observed for RESIT on PNL data. The violin plot of the FNR-pi on non-iid data completely overlaps with that on vanilla data, and there is only a slight shift in the median (< 0.05), thus we do not think it is of scientific relevance as it might be due to the samples fluctuations: given that evaluation is done on newly sampled data for each scenario, we are not interested in absolute statements about each sample, but instead about statements that appear to be statistically significant with respect to our experiments.

---

> > ### Comment · Reviewer_eJF4 · 2023-08-21
> >
> > Thanks to the authors for their careful reading of my review and their detailed replies. Between the overall response and the specific response above, I'm satisfied that many of my concerns have been addressed. In particular, I'm expecting that the authors will make good on their offers to include additional discussions, either in the main body of the paper or the supplementary materials (with a strong preference for the former). For example, the rough equivalence of results when using BSF should be mentioned in the body of the paper and then results provided in the supplemental materials.
> >
> > I'll increase my score.

---

### Official Review · Reviewer_QK5D · 2023-07-12

**Soundness:** 2 fair
**Presentation:** 3 good
**Contribution:** 2 fair
**Rating:** 3
**Confidence:** 4

**Summary:**

The paper considers the important problem of what happens to the reliability of the causal model output of different types of causal discovery algorithms when the underlying assumptions behind the method are violated. It finds that score matching-based methods tend to be more robust and provides a theoretical explanation.


**Strengths:**

- very important and challenging problem that is often encountered in practice
- good overview of causal discovery methods and common assumptions behind them
- highly commendable attempt at providing a large suite of software + data sets for future standardised benchmark evaluations on (new) causal discovery algorithms


**Weaknesses:**

- Unfortunately, despite the laudable goal, the comparison is too limited and biased to be meaningful.
The generated models all have nonlinear interactions and/or non-Gaussian noise … which is a key assumption for e.g. score match-based methods. But then you should also see how they hold up in the standard linear Gaussian case (answer: terrible) before you start proclaiming their ‘remarkable robustness’.
Similarly, if ‘additive noise’ is key, then compare against e.g. multiplicative noise (not ‘post nonlinear’, as that can often be treated in the original way). ‘Faithfulness’ relates to independence based methods (CAM/PC/GES), so ‘unfaithful’ is not a misspecified model for the score match-based algorithms … so hardly ‘remarkable’ they are unaffected by that scenario. No consideration of acyclic vs. cyclic (in practice nearly all relevant systems have feedback mechanisms).

- trying to get to a universal claim / quantification on what is ‘the’ most robust method is doomed to failure as it is an ill-posed problem. Instead what you can meaningfully do is try to assess how the performance of individual methods degrades as the degree of violation of assumptions increases: is there graceful degradation or sudden collapse beyond a certain threshold? Is a method more sensitive to some types of violations than others? Is the entire model ‘bad’, or is the skeleton still reliable and mainly the orientations that suffer? etc.

- it is bad practice to take the outcome of a single run of a causal discovery algorithm as ‘the’ answer. At least one should do a form of bootstrapping to come to a balanced model interpretation, and use that as the basis for comparison between methods. Also, the FNR-\pi metric used in the evaluation (Fig1(a-f)-right) is hard to interpret as a measure of model quality: instead something like the structural Hamming distance may be more appropriate.

- main claim that ‘score matching-based methods are most robust’ is not justified based on the evidence presented in Fig.1 (see remarks below)

- reported experimental results of some methods (PC/GES) are so laughably bad (much worse than random guessing) that I have no idea what happened, but something went wrong


**Questions:**

(see also remarks above)
- why do you try to emphasise the claim that ‘score matching procedures are *robust*’ whereas if anything in Fig.1 the FNR-\pi performance of RESIT is almost always nearly identical between the vanilla and ‘misspecified’ cases, contrary to the score matching algorithms that do often show a drop in FNR-\pi? (so RESIT may not be the ‘best’, but it is the most *robust* one)

Other remarks (for lack of a better place):
- typo title ‘casual’
- 28: ‘These strong conditions are arguably necessary’ => No: there are a whole range of algorithms that can handle confounders (or even cycles) these days, and the FCI benchmark algorithm has been around for over 30 years now.
- 57: in one sentence you mention 12 papers that are irrelevant to yours ….
- Table 1: GES optimises the Gaussian likelihood => no: GES is a search strategy. It incorporates a scoring metric, but that does not need to be the Gaussian likelihood. (Although it might explain the observed performance).
- 203: ‘no straightforward comparison between PAGs and CPDAGs’ => ?? both show edges and edge marks, and indicate when they don’t know …
- 214: ‘differences .. mostly in the FNR-\pi score’ => I have no idea why that should be the case, especially since you say all start from the same skeleton procedure? Also in FIg.1 RESIT shows no change in FNR-\pi even though it does show significant differences in F1. Most importantly, however, is that it is very difficult as a reader to interpret this score in terms of how ‘good’ or ‘robust’ the actual causal model output is.

---

> ### Author Rebuttal · Authors · 2023-08-09
>
> We thank the reviewer for the valuable insights provided with their comment. In the response, we will follow the structure of the bullet list in the review.
>
> -   The reviewer comments that our "comparison is too limited and biased to be meaningful”, supporting their claim with a series of points.
>
> 	> The generated models all have nonlinear interactions and/or non-Gaussian noise …
>
> 	Our models include:
>
> 	1. Linear mechanisms
>
> 	2. ANM with nonlinear mechanisms
>
> 	3. PNL model with nonlinear mechanisms
>
> 	The choice of the noise terms is motivated by the fact that the distinction between Gaussian and non-Gaussian is what really matters in the assumptions of the methods involved, with CAM, SCORE, DAS, GraN-DAG, DiffAN only supporting Gaussian noise, while RESIT, NoGAM, and PC supporting both Gaussian and non-Gaussian noise, and DirectLiNGAM supporting only non-Gaussian noise terms. Given that these are the distinctions to bear in mind, our choice is consistent with the assumption we are willing to probe (i.e. effect of misspecification of the noise distribution on the performance).
>
> 	> You should also see how they hold up in the standard linear Gaussian case (answer: terrible)
>
> 	As correctly remarked, there is no reason to expect that methods based on finding the topological ordering will be robust in the setting where ordering can not be identified for a pair of nodes (see L 98-99).
>
> 	> … ‘unfaithful’ is not a misspecified model ...
>
> 	We agree with the reviewer, which is why we only define as “_surprising_” the “_decrease in FNR-pi performance with respect to the vanilla scenario_” (L 304-305), and immediately after we explain the observed results, connecting graph sparsity with the observed performance (L309-311). As noticed by reviewer eJF4, this is in line with previous literature: [1] Costantinou et al., 2019 point out that graph sparsity directly affects the f1 score. Our observation that associates unfaithful path canceling with this fact from previous literature is thus an original, pertinent and meaningful statement.
>
> 	We are happy to add the citation to [1] to back our claim and to remove the word _surprising_ from the main text, to enhance better clarity.
>
> 	[1] _Evaluating structure learning algorithms with a balanced scoring function_, Costantinou et al., 2019
>
> 	> _No consideration of acyclic vs. cyclic_
>
> 	Cyclic graphs are beyond the scope of this paper. We explicitly rely on topological ordering for evaluating the graphs (FNR-pi metric). This is clearly not possible with cyclic graphs. Moreover, despite the fact that in principle one would like to test any possible violation, this is not possible in the scope of a single paper, where our work already tackles important questions in a large scale effort, as recognized by the reviewers (”highly commendable attempt at providing a large suite of software + data sets”, R QK5D; “experiments are extensive.”, R eJF4).
>
> -   > trying to get to a universal claim/quantification on what is ‘the’ most robust method is doomed to failure as it is an ill-posed problem. ...: is there graceful degradation or sudden collapse beyond a certain threshold?
>
>
> 	In this regard, we note that Appendix B.5 provides a detailed discussion on the parametrization of the models with respect to the _amount_ of violation. In Appendix B.5 we also clarify that the results are presented in the setting where the parameter maximizes the amount of violation, e.g. for measurements error, we specify that we show results for the maximal noise-signal ratio: with the measurement error example in mind, we argue that if we observe robust performance of a method in this _extreme_ setting maximizing the noise-signal ratio, we expect good/better performance to be observed when noise-signal ratio is reduced, given that the distribution of the data would get closer to that of data generated by the vanilla model. Given that finer grained analysis of the experiments is not compatible with space limits, we will add a section discussing results degradation with respect to parameter changes, as we agree with the reviewer that this would be a valuable contribution.
>
> 	> _Is a method more sensitive to some types of violations than others?_
>
> 	This information is already in the paper, and it is actually one of its key contributions. Section 4.1 is indeed structured in paragraphs, each one analyzing which methods are more sensitive and which are more robust (in the sense of performance under model misspecification) with respect to the assumption violation introduced in a specific data generating process. Similarly, Figure 1 (a-f) provides a comprehensive overview of the effect of each type of violation on the performance of each algorithm.
>
> 	> _Is the entire model ‘bad’, or is the skeleton still reliable and mainly the orientations that suffer?_
>
> 	We agree that this is valuable information, and we commit to add such evaluation in the appendix of the paper. We add preliminary plots in the pdf attached to the rebuttal in the common response (Figure 1.)
>
> -   > it is bad practice to take the outcome of a single run of a causal discovery algorithm as ‘the’ answer ...
>
>
> 	We agree that when dealing with real data in applications, the standard practice is to use some type of bootstrap to quantify the uncertainty on edges. On the other hand, in the case of algorithms' evaluation and comparison, it is a well established and the most common practice in literature to consider the outcome of a single run as the answer (see, for instance, the papers of SCORE, DAS, NoGAM, GraNDAG, CAM, to cite a few of those that we used in our work). This difference is motivated by the fact that in practical applications the user is interested in inference on a single graph. In our setting, instead, adding bootstrapping on top of 2M experiments is clearly not a feasible option.
>
> -   For the last two points, please refer to the common answer.

---

### Author Rebuttal · Authors · 2023-08-09

We thank the reviewers for their effort in reading our paper also providing valuable insights. In general, there is agreement that this is _very important_ work (R QK5D , eJF4, JGO2), both in terms of the subject of the paper and in the scale of the experiments (_highly commendable attempt at providing a large suite of software + data sets for future standardized benchmark evaluations_, R QK5D; _experiments are extensive_, R eJF4), with meaningful practical impact. Moreover, there is appreciation in the presentation of the results, with the use of a random baseline that _is a great choice and helps readability_ (R eJF4, R JGO2), and in the choice of the tested violations (R JGO2).

In this common response, we address comments shared by more than one reviewer. We are responding to each reviewer independently on the remaining points.

-   R QK5D expresses concerns with respect to the performance of PC and GES methods with respect to the random baseline, mentioning “_reported experimental results of some methods (PC/GES) are so laughably bad (much worse than random guessing) that I have no idea what happened, but something went wrong_”, whereas R eJF4 suggests that the paper should explain why PC generally performs worse than random in terms of F1.

	We respectfully disagree with the comment from R QK5D that “_something went wrong_”. Figures from 14 to 18 illustrate the methods’ performance (FNR and FPR) on ER graphs with 5 and 10 nodes, both sparse and dense, and on sparse graphs of 20 nodes: under these conditions, PC and GES perform consistently better than random in the vanilla case, whereas for the misspecified scenarios, we observe performance reasonably close to the remainder of the benchmarked methods. Additionally, we remark (as already reported in the paper, footnote 1 of page 6) that we used PC and GES implementation from the well-established and maintained library _DoDiscover__._ Moreover, we report that our findings are in line with those in the literature: for example, [1, 2, 3, 4, 5, 6] all explicitly state that PC and/or GES are excluded as baselines for the evaluation of their proposed method, as they perform “_significatly worse_”. In light of these considerations, we have no reason to believe that something went wrong or that there are bugs in the code.

	Finally, we provide a more detailed explanation of the observed performance of PC and GES. In the dense and large graphs case (which is the one under consideration in Figure 1), a lot of quasi-violations of faithfulness are expected to happen (see _Geometry of the faithfulness assumption in causal inference_, Uhler et al., 2012 ). Thus, we would expect in this setting PC and GES to be systematically wrong (which is what we observe), and coherently to this, we do not expect any significant drop in performance of these two methods switching from the vanilla to the unfaithful setting (which is coherent with the observations of Figure 1e). If the reviewers agree that this is a valuable point, we are willing to include this discussion in Section 4, in order to provide better clarity with respect to the observed results for PC and GES.

-   R QK5D and R eJF4 express concern about our claim that score matching-based methods provides robust performance with respect to model misspecification. In relation to this, R QK5D adds ”_if anything in Fig.1 the FNR-\pi performance of RESIT is almost always nearly identical between the vanilla and ‘misspecified’ cases, contrary to the score matching algorithms that do often show a drop in FNR-\pi? (so RESIT may not be the ‘best’, but it is the most robust one)”_. We merge our response to the two points (score match and RESIT robustness) here, as these two points are related.

	In the case where the vanilla performance is comparable to the random baseline (which is the case for RESIT, Fig1 a-f) the method is not only _not the best_ (as remarked by the reviewer in reference to RESIT), but in general unable to make inference better than random with statistical significance in _any_ of the settings. If we don’t account for this systematic inability of the method of doing reliable inference, we should conclude that the random baseline is the most robust causal discovery algorithm under assumption violation, but this would clearly be an ill-defined notion of robustness.

	On the other hand, score matching-based methods are the only ones that show performance better than random with statistical significance across almost all of the considered data generating scenarios. In this regard, we point out that in machine learning robustness is generally intended as the ability of the model to perform well in misspecified scenarios. If the use of the word _robustness_ sounds confusing in this setting, we are happy to replace it with the expression _performance under misspecification_. In this sense, the best performance is consistently provided by score matching-based methods, especially in terms of their ability to infer the topological ordering (measured by FNR-pi score).

_References_

[1] Score matching enables causal discovery of nonlinear additive noise models, Rolland et al., 2022

[2] Causal discovery with score matching on additive models with arbitrary noise, Montagna et al., 2023

[3] Scalable causal discovery with score matching, Montagna et al., 2023

[4] CAM: Causal additive models, high-dimensional order search and penalized regression, Buhlmann et al., 2014

[5] Gradient-based neural dag learning, Lachapelle et al., 2019

[6] Dags with no tears: Continuous optimization for structure learning, Zheng et al., 2018

---

### Decision · Program_Chairs · 2023-09-21

**Decision:**

Accept (poster)

**Comment:**

Two of the reviewers recommend acceptance and were further strengthened in their recommendations by the authors' replies.  One reviewer disagrees, so we could not reach consensus.  I am recommending acceptance based on all the comments taken into consideration in total, but the comments of that last reviewer should be addressed in the final paper by the authors.  Of particular importance are the comments in by that reviewer in reply to the author rebuttal, reproduced here:

"I am afraid my main concern still stands: laudable goal, ok method, but overstated claims based on biased evaluations. The method relies strongly on nonlinearity of interactions or non-Gaussianity of the noise. Other methods can have other assumptions (e.g. 'linear Gaussian'). If you want to claim your method is much more robust than others, you cannot present a range of scenarios where each case satisfies your key signal for inference (all scenarios involve strong nonlinearities or strongly non-Gaussian noise), but violates those of one or more others. To then claim 'look how much more robust our method is in general' is then at best 'not sufficiently supported'.

The authors acknowledge the method will perform terrible in the linear Gaussian setting. So then show how the performance behaves in systems that vary in degrees of nonlinearity + non-Gaussianity. Does the algorithm still perform well in small to moderate amounts of nonlinearity + non-Gaussianity or not? How strong do they need to be? What if parts of the model have interactions that are close to 'linear Gaussian'? etc.

As a result, I think the claims in the paper could be potentially misleading to users that are not aware of these restrictions, and therefore I am afraid I have to stick to reject."